# Continuous measurements of isotopic composition of water vapour on the East Antarctic Plateau

Mathieu Casado[1,2], Amaelle Landais[1], Valérie Masson-Delmotte[1], Christophe Genthon[4,5], Erik Kerstel[2,3], Samir Kassi[2], Laurent Arnaud[4,5], Ghislain Picard[4,5], Frederic Prie[1], Olivier Cattani[1], Hans-Christian Steen-Larsen[6], Etienne Vignon[4,5], Peter Cermak[7]

[1.] Laboratoire des Sciences du Climat et de l'Environnement - IPSL, UMR 8212, CEA-CNRS-UVSQ, Gif sur Yvette, France

[2.] CNRS, LIPHY, F-38000 Grenoble, France

[3.] Univ. Grenoble Alpes, LIPHY, F-38000 Grenoble, France

[4.] Univ. Grenoble Alpes, LGGE, F-38041 Grenoble, France

[5.] CNRS, LGGE, F-38041 Grenoble, France

[6.] Centre for Ice and Climate, University of Copenhagen, Denmark

[7.] Department of Experimental Physics, Faculty of Mathematics, Physics and Informatics, Comenius University, Mlynska dolina F2, 842 48 Bratislava, Slovakia

*Correspondence to*: mathieu.casado@gmail.com

**Abstract.** Water stable isotopes in central Antarctic ice cores are critical to quantify past temperature changes. Accurate temperature reconstructions require to understand the processes controlling surface snow isotopic composition. Isotopic fractionation processes occurring in the atmosphere and controlling snowfall isotopic composition are well understood theoretically and implemented in atmospheric models. However, post-deposition processes are poorly documented and understood. To quantitatively interpret the isotopic composition of water archived in ice cores, it is thus essential to study the continuum between surface water vapour, precipitation, surface snow and buried snow.

Here, we target the isotopic composition of water vapour at Concordia Station, where the oldest EPICA Dome C ice cores have been retrieved. While snowfall and surface snow sampling is routinely performed, accurate measurements of surface water vapour are challenging in such cold and dry conditions. New developments in infrared spectroscopy enable now the measurement of isotopic composition in water vapour traces. Two infrared spectrometers have been deployed at Concordia, allowing continuous, in situ measurements for one month in December-January 2014-2015. Comparison of the results from infrared spectroscopy with laboratory measurements of discrete samples trapped using cryogenic sampling validates the relevance of the method to measure isotopic composition in dry conditions. We observe very large diurnal cycles in isotopic composition well correlated with temperature diurnal cycles. Identification of different behaviours of isotopic composition in the water vapour associated with turbulent or stratified regime indicates a strong impact of meteorological processes in local vapour/snow interaction. Even if the vapour isotopic composition seems to be, at least part of the time, at equilibrium with the local snow, the slope of $\delta D$ against $\delta^{18}O$ prevents us from identifying a unique origin leading to this isotopic composition.

## 1. Introduction

Ice cores from polar ice sheets provide exceptional archives of past variations in climate, aerosols, and global atmospheric composition. Amongst the various measurements performed in ice cores, the stable isotopic composition of water (e.g. $\delta^{18}O$ or $\delta D$) provides key insights in past polar climate and atmospheric water cycle. The atmospheric processes controlling this signal have been explored throughout the past decades using present day monitoring data. Based on the sampling of precipitation or surface snow, relationships between precipitation isotopic composition and local temperature have been identified since the 1960s, and understood theoretically to reflect atmospheric distillation processes (Dansgaard, 1964; Lorius et al., 1969). Nevertheless, there is both observational and modelling evidence that the isotope-temperature relationship is not stable in time and space (Jouzel et al., 1997; Masson-Delmotte et al., 2008). The variation in the isotope-temperature relationship have been explained by the isotopic composition of precipitation being sensitive to changes in condensation versus surface temperature, to changes in evaporation condition and transport paths, and to changes in precipitation intermittency (Charles et al., 1994; Fawcett et al., 1997; Krinner et al., 1997; LeGrande and Schmidt, 2006; Masson-Delmotte et al., 2011; Werner et al., 2011). While complex, these processes can be tracked using second order isotopic parameters such as d-excess, which preserve information on evaporation conditions (Jouzel et al., 2013; Landais et al., 2008) and they are accounted for by atmospheric models equipped with water stable isotopes (Risi et al., 2010; Schmidt et al., 2005; Werner et al., 2011).

The variations of d-excess and some variations in $\delta^{18}O$ are due to the different influences of equilibrium fractionation and diffusion driven kinetic fractionation processes at each step of the water mass distillation trajectory. Specific limitations exist for the representation of the isotopic fractionation at very low temperature. Equilibrium fractionation coefficients have been either determined by spectroscopic calculations (Van Hook, 1968) or by laboratory experiments (Ellehøj et al., 2013; Majoube, 1971; Merlivat and Nief, 1967) with significant discrepancies at low temperatures. Molecular diffusivities have mainly been measured at 20°C (Cappa et al., 2003; Merlivat, 1978) but recent experiments have shown that temperature can have a strong impact on these coefficients (Luz et al., 2009).

Another source of uncertainty for the climatic interpretation of ice core records arises from poorly understood post-deposition processes. Indeed, the isotopic signal of initial local snowfall can be altered through wind transport and erosion, which are strongly dependent on local and regional topography, and can produce artificial variations in ice core water stable isotopes caused by gradual snow dune movement (Ekaykin et al., 2002; Ekaykin et al., 2004; Frezzotti et al., 2002). Moreover, it is well known that the initial isotopic signal associated with individual snowfall events is smoothed in firn, a process described as "diffusion" (Johnsen et al., 2000; Neumann and Waddington, 2004). This "diffusion" occurs through isotopic exchanges between surface water vapour and snow crystals during snow metamorphism (Waddington et al., 2002). "Diffusion lengths" have been identified based on spectral properties of ice core records and shown to depend on several processes: wind transport and erosion will alter the surface composition with a very strong influence of orography, diffusion through the pores of the snow firn smooths the signal as does metamorphism of the crystals (Schneebeli and Sokratov, 2004). Finally, there are hints based on high resolution isotopic measurements performed near snow surface for potential alteration of the initial precipitation isotopic composition (Hoshina et al., 2014; Sokratov and Golubev, 2009; Steen-Larsen et al., 2014a). This motivates investigations of the isotopic composition of precipitation, surface snow, but also of surface water vapour.

Atmospheric monitoring in extreme polar climatic conditions remains challenging. Supersaturation generates frost deposition which can bias temperature and humidity measurements, and low vapour contents are often outside of range of commercial instruments. As specific humidity is under 1000 ppmv on the central Antarctic plateau, measuring the isotopic composition of surface water vapour requires either very long cryogenic trapping (typically 10 hours at 20L/min) to collect enough material for off-line (mass spectrometric or laser-based) isotopic analyses, or very sensitive on-line (laser-based) instruments able to produce accurate in-situ isotopic measurements.

Recent developments in infrared spectroscopy now enable direct measurements of isotopic composition of the vapour in the field, without time-consuming vapour trapping. With careful calibration methodologies, these devices provide accuracies comparable with those of mass spectrometers (Bailey et al., 2015; Tremoy et al., 2011) and have already been used for surface studies in the Arctic Region (Bonne et al., 2015; Bonne et al., 2014; Steen-Larsen et al., 2014a).

The goal of our study is first to demonstrate the capability to reliably measure the isotopic composition of central Antarctic surface water vapour during summer, second to investigate the magnitude of its diurnal variations, in comparison with the corresponding results from central Greenland (Steen-Larsen et al., 2013), and third to highlight the impact of a intermittently turbulent boundary layer on the isotopic composition variations.

We focus on Concordia station, at the Dome C site, where the oldest Antarctic ice core record, spanning the last 800 000 years, has been obtained (EPICA, 2004). During the last 20 years, the French-Italian Concordia station has been progressively equipped with a variety of meteorological monitoring tools, documenting vertical and temporal variations in atmospheric water vapour (Ricaud et al., 2012). During summer, meteorological data depict large diurnal cycles in both surface air temperature and humidity (Genthon et al., 2013), which may result from either boundary layer dynamics and/or air-snow sublimation/condensation exchanges.

During the Antarctic summer of 2006-2007, cold trap samplings of water vapour were performed. Here, we report for the first time the results of this preliminary study together with continuous measurements performed during the austral summer of 2014-2015 using laser instruments with a specific methodology for low humidity calibration, as well as new cold trap sampling for laboratory measurements.

This manuscript is organised in two main sections to highlight the two different aspects of the study. First, section 2 describes the technical aspect: the site, the material deployed and the applied methods, with a focus on calibration in order to assess the technical reliability of such methods for sites as cold as the Antarctic Plateau. Section 3 reports the scientific aspect of the results, with first a focus on the relevance of infrared spectroscopy compared to cryogenic trapping, second a description of the diurnal to intra-seasonal surface vapour isotopic variations and third an analysis of the origin of the vapour. We conclude and discuss outlooks for this work in Section 4.

## 2.    Technical challenges

### 2.1.    Sampling site

Concordia station is located near the top of Dome C at 75°06'06"S - 123°23'43"E, 3233 m above sea level and 950 km from the coast. While the local mean temperature is -54.3°C, it was -32.4°C during the campaign of 2014/2015 reaching a maximum value of -24.5°C. Ice core data suggest an average annual accumulation of 2.7±0.7 g.cm$^{-2}$.yr$^{-1}$ (Genthon et al., 2015; Petit et al., 1982; Röthlisberger et al., 2000).

The first cold trap vapour sampling campaign was performed in summer 2006-2007. The second field campaign took place from December the 24th 2014 to January the 17th 2015.

The spectrometers for the 2014/2015 campaign were installed in an underground shelter located 800m upwind from the station, therefore protected from the fumes of the power generator of the station (discussed in section 2.5). Such underground shelter allows us to avoid any impact of the monitoring structure on the wind field and possible sampling artefacts. The area around the shelter is characterized by few sastrugi, none

higher than 20cm (Figure 1). A clean area of 12m² with no sastrugi was marked around the inlets. We decided to point the inlets toward the dominant wind in order to prevent artefacts from condensation or evaporation from the protection of the inlet or the pole holding it. Indeed, frost formation was observed on the protective foam and pole.

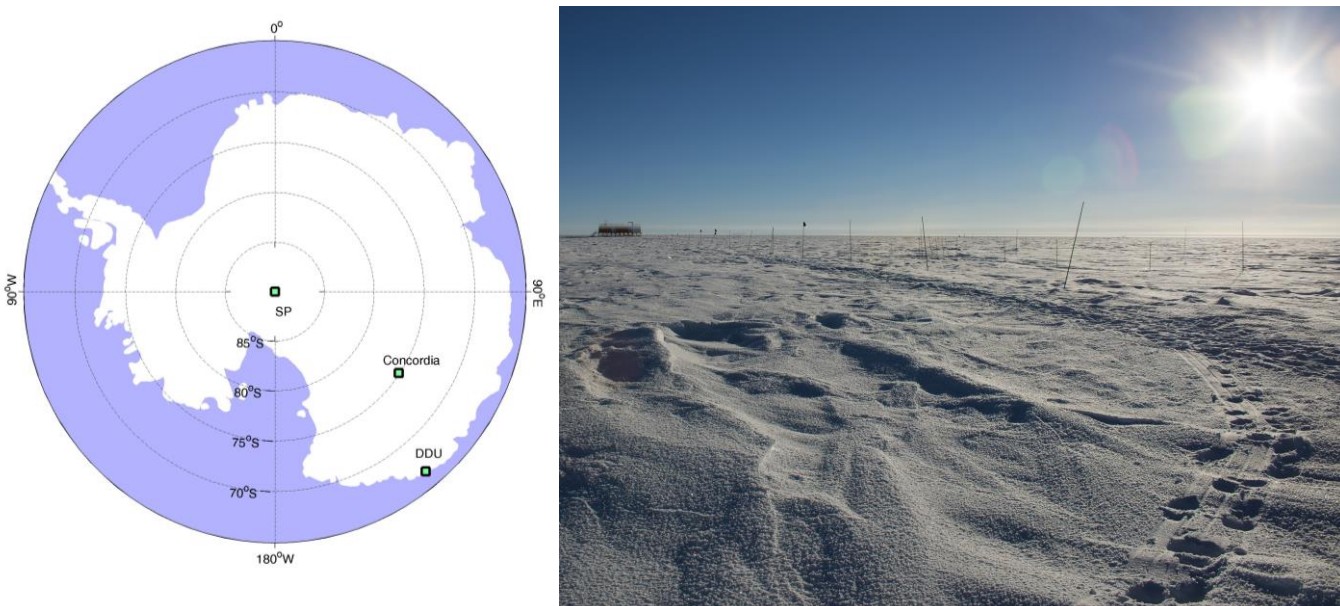

*Figure 1: Map of Antarctica showing the location of Concordia, Dumont d'Urville station (DDU) and the South Pole (SP). Picture of the area from the top of the underground shelter where the instrument was located looking toward the clean area.*

Together with our water vapour isotopic data, we use meteorological observations from the lowest level of the 45m meteorological profiling system at Dome C (Genthon et al., 2013). The profiling system was located at proximity with the spectrometers. The temperature observations on the 45m profiling system are made in aspirated shields and thus not affected by radiation biases. Genthon et al. (2011) demonstrated that when the wind speed is below 5m.s$^{-1}$, radiation biases are very significant and can reach more than 10°C in conventional (non-wind ventilated) shields. Temperature is measured using HMP155 thermohygrometers, wind speed and direction using Young 05103 and 05106 aerovanes. Elevation above the snow surface was 3.10m for the wind and 2.58m for temperature in 2014-15. This will be henceforth commonly referred as the 3m level. Further details on the observing system, instruments, sampling and results are available in previous publications (Genthon et al., 2013; Genthon et al., 2010). Surface temperature is measured with a Campbell scientific IR120 infrared probe. The probe is located at 2 meters height and uses upwelling infrared radiation and the temperature of the detector to compute the temperature of the surface of the snow. The uncertainty of the surface temperature measurement is around ±1°C which is mainly due to unknown and possibly varying emissivity of the snow (Salisbury et al., 1994).

## 2.2. Water vapour isotope monitoring

Two infrared spectrometers were used to measure continuously the isotopic composition of water vapour pumped 2 meters above the snow surface: a Cavity Ring-Down Spectrometer (CRDS) from Picarro (L2130-i) and a High-Finesse water Isotope spectrometer (HiFI) based on the technique of optical feedback cavity enhanced absorption spectroscopy (OFCEAS) developed in LIPhy, Grenoble, France (Landsberg et al., 2014) as described on Figure 2.

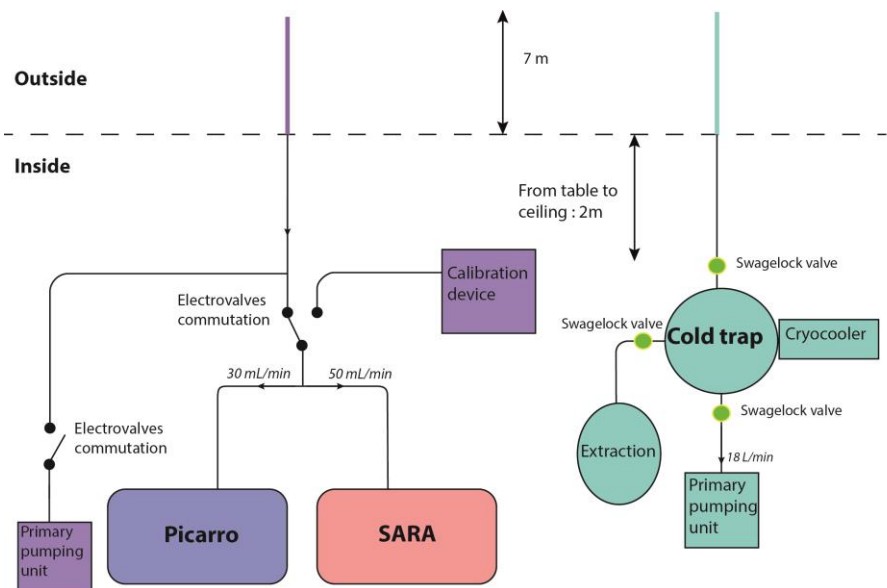

*Figure 2: Schematics of the experimental set up implied in the water vapour isotopic composition monitoring*

Both instruments are based on a general technique of cavity enhanced absorption spectroscopy (Romanini et al., 2014). This is essentially a long path length optical detection technique that increases the sensitivity of detection of molecules in the optical cavity by folding the optical beam path between two (or three) highly
reflective mirrors. The commercial Picarro spectrometer is based on near-infrared continuous-wave cavity ring-down spectroscopy (CW-CRDS) (Crosson, 2008). It has proven to be a fairly robust and reliable system, delivering good precision isotopic measurements at concentration (water mixing ratio) values between 1000 and 25000 ppmv.

The HiFI spectrometer is also operating in the near-infrared region of the spectrum, but uses another implementation of cavity enhanced absorption spectroscopy called optical feedback cavity enhanced absorption spectroscopy (OFCEAS) (Romanini et al., 2014). In the case of the HiFI spectrometer, the optical path length was increased by about one order of magnitude to 45 km. This optimizes the spectrometer for oxygen-18 isotopic measurements with a precision better than 0.05‰ at a water mixing ratio around 500
ppmv (Landsberg et al., 2014). The HiFI spectrometer was shown to be able to reach this level of performance also in Antarctica during a 3-week campaign at the Norwegian station of Troll (Landsberg, 2014). Unfortunately, during the current campaign at Dome C, the spectrometer had to operate in a noisy environment. The system was not isolated from vibrations of several vacuum pumps in the shelter and an accidental resonance did perturb the phase control. This resulted in a baseline noise level more than one order
higher than normal, which created a corresponding increase of the error on the isotope ratio measurements. At this level of noise, the Picarro measurements turned out to be more precise than the HiFI measurements. It is for this reason that the latter were only used as an independent tool to check on the absolute values from Picarro measurements. All time-series shown hereafter were obtained with the Picarro spectrometer.

The two instruments were connected through a common heated inlet consisting of a ¼ inch copper tube. The internal pumps of each instrument pumped the outside vapour through the common inlet and into the respective cavities. The fluxes generated by the instruments were small enough not to interact with one another, as attested by stable pressure in the cavities of both instruments. The length of this common inlet (approximately 10m long) caused a response delay of approximately 2 minutes for the humidity signal.
Memory effects caused by interactions between the water vapour and the inside of the tubes introduce different delays for different isotopes. In the case of high resolution data, artificial d-excess can be produced as the memory effect of HDO is substantial larger than $H_2^{18}O$ (Steen-Larsen et al., 2014b). However our

measurements were averaged over 1 hour thereby removing this effect. No sign of condensation in the inlet was observed during the whole campaign.


### 2.3. Allan variance analyses

The measurements of isotopic composition with an acquisition time of approximately 1 second have a standard deviation of 10‰ for $\delta D$ and of 2‰ for $\delta^{18}O$ at approximately 500 ppmv (Figure 3). Infrared
spectrometers typically produce data perturbed by different kinds of noise: one is noise due to frequency instabilities of the laser, temperature and mechanical instabilities of the cavity, temperature and pressure of the sampled gas, electronic noise, and residual optical interference fringes on the spectrum baseline. These noise, usually predominantly white noise, can be significantly reduced through time averaging: for instance, with an acquisition time of 2 minutes, we decrease the standard deviation to 1.3‰ for $\delta D$ and 0.2‰ for $\delta^{18}O$.

With increasing integration time, one expects the precision of the measurements to initially improve, due to the reduction of white noise, up to the point where instrumental drift becomes visible. The so-called Allan-Werle plot shows the overall expected precision as a function of integration time (Figure 3).

Laboratory long term measurement of a standard was carried out at a humidity of 506 ± 3 ppmv in order to reproduce the range of the expected humidity for Concordia station. Stable humidity production for 13 hours was realised using the calibration device described in the next section and in the supplementary material 1. The standard deviation of $\delta D$ follows the optimum line almost up to 4 hours integration time. The standard deviation of $\delta^{18}O$ does not follow the optimum profile after 100 seconds but still drops continuously over
almost 2 hours. These measurements confirm the reliability of the Picarro L2130i even at low humidity and justify the use of such an instrument in this campaign. The integration time providing the ultimate precision could not be achieved because of the lack of a vapour generator stable for more than 13 hours. At other humidity levels, we observe similar profiles with an increasing initial precision as the moisture content increases (not shown).

In the field, we performed calibrations lasting up to 90 minutes, as a trade-off between instrument characterization and measurement time optimization. This, however, is not long enough to accurately estimate the rise of uncertainty due to instrumental drift, but allows us to assess the ultimate precision for the instruments under realistic field conditions. The Allan variance was thus calculated from field Picarro
calibration data, at 450 ppmv. From this analysis, we conclude that 2 minutes appear to provide an optimal integration time, associated with an ultimate precision of the spectrometer of 0.2‰ for $\delta^{18}O$ and 1.1‰ for $\delta D$ (black dashed lines on Figure 3). This test could not be performed at other humidities.

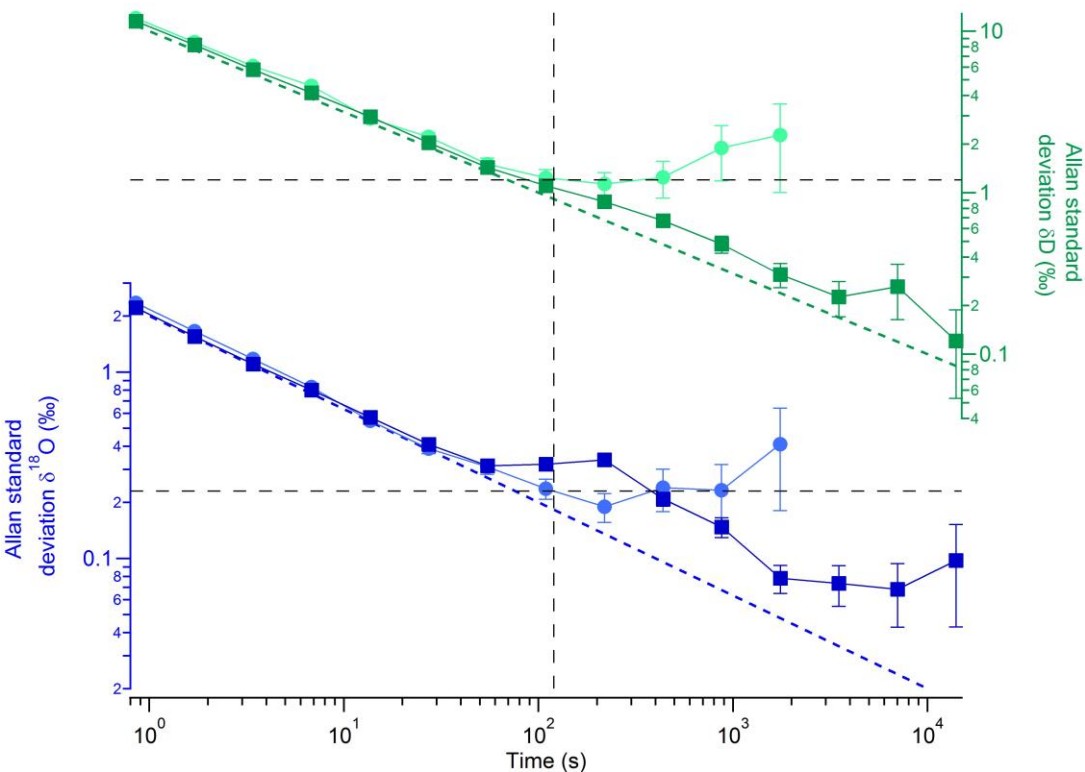

*Figure 3: Allan variance plots for laboratory long term standard measurement (dark squares) and for field long term standard measurement (light circles) for δD (Top, green) and for δ<sup>18</sup>O (bottom, blue). Dash lines correspond to the quantum limit on $N^{-1/2}$ for each composition.*

## 2.4. Calibrations

Calibration of the spectrometer is crucial in order to be able to express the measurement results with confidence on the international VSMOW2-SLAP2 isotope scale (IAEA, 2009). Calibrations have been reported to vary between instruments and calibration systems, as well as over time. Tremoy et al. (2011) highlighted the importance of calibration for Picarro analysers under 10 000 ppmv with biases up to 10‰ for $\delta D$ and of 1‰ for $\delta^{18}O$ at volume mixing ratios (VMR) down to 2000 ppmv. Protocols have been developed and adapted for calibration under Greenland ice sheet summer (Steen-Larsen et al., 2013) and south Greenland year-round conditions (Bonne et al., 2014) with good performance attested from parallel measurements of PICARRO and LGR analysers for humidity above 2000 ppmv. At VMRs below 2000 ppmv, much larger errors can be expected without calibrating the instruments.

For this field season, we have followed the classical calibration protocols with (1) a study of the drift of the instrument, (2) a linearity calibration using two working standards whose isotopic values were established in the laboratory versus SMOW and SLAP and (3) a study of the influence of humidity on the isotopic value of the water vapour. At very low humidity levels (below 2000 ppmv), standard calibration devices (such as the SDM from Picarro) are not able to generate stable constant humidity. Here, we expected humidity levels below 1000 ppmv and therefore we could not use standard water vapour generator and had to develop our own device inspired from the device developed by Landsberg (2014) and described in detail in the supplementary materials section 1.

The calibration protocol for type (1) calibration relies on the measurement of 1 standard at 1 humidity level (the average of the expected measurement) twice a day for 30 minutes in order to evaluate the mean drift of the infrared spectrometer. Standard values of the drift on a daily basis should not exceed 0.3‰ in $\delta^{18}O$ and

2‰ in δD. The calibration protocol for type (2) calibration relies on the measurement of 2 standards whose isotopic compositions bracket the one measured in order to evaluate the response of the infrared spectrometer compared to the SMOW-SLAP scale (thereafter isotope-isotope response). Typical isotope-isotope slope is between 0.95‰/‰ and 1.05‰/‰ for $\delta^{18}O$ and for δD. The calibration protocol for type (3) calibration relies on the measurements of 1 standard at different levels of humidity in order to evaluate the response of the infrared spectrometer to humidity (thereafter isotope-humidity response). Type (2) and type (3) calibration can only be realised once a week provided type (1) calibration has validated the drift of the instrument was within acceptable values (below excess 0.3‰ in $\delta^{18}O$ and 2‰ in δD). For temperate range where humidity is important (above 5000 ppmv), it is possible to consider a linear relationship for the isotope-humidity response; for dryer situations (below 5000 ppmv), the isotope-humidity response requires at least a quadratic relationship.

The 3 types of calibrations were performed in the field and in the laboratory prior and after field work. It was particularly important to add laboratory calibrations (especially for drift of the instrument) in addition to field calibrations because of the short season and lack of dry air at the beginning of the season, in particular to strengthen the results from type (2) and (3) calibrations as we will present in the following.

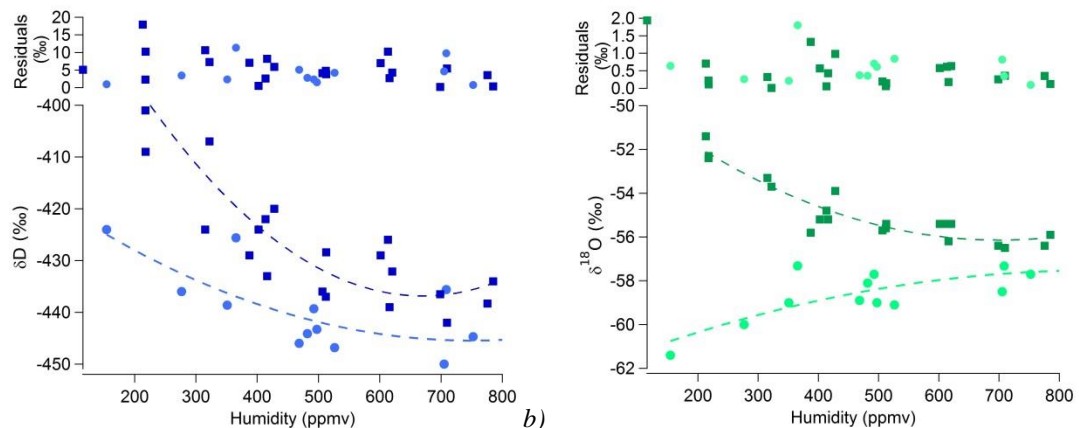

_Figure 4: Measured isotopic composition for a) δD and b) $\delta^{18}O$ using the PICARRO spectrometer for a fixed humidity: light circles are field calibration points, dark squares are laboratory calibration points, the dashed lines are the fit with a quadratic function and on top are the residuals compared to the fit for the entire series._

In order to evaluate the performances of our spectrometer, all type of calibrations were performed in the laboratory at different humidities (from 100 to 1000 ppmv) and repeated on five occasions in a time span of 4 weeks with two standards UL1 : $\delta^{18}O$ = -54.30‰ and δD= -431.1‰ and NEEM : $\delta^{18}O$ = -33.56‰ and δD= -257.6‰. We estimate the mean drift for a period of a month (type (1)) by comparing the offset of the isotopic composition over the 5 occurrences. For the isotope-isotope slope, we obtain standard values around 0.95‰/‰. We evaluate the laboratory isotope-humidity response by comparing the measured value of the isotopic composition to the value of humidity. Each independent set of calibrations (each week) can be fitted by a quadratic function with a small dispersion of the data points (inferior to 2‰ for δD and 0.2 ‰ for $\delta^{18}O$). Different calibration sets performed over different days show dispersion due to the instrument drift. We observe a much larger dispersion for δD than for $\delta^{18}O$, in particular at low concentration (200 ppmv) due to the combined action of the drift and of the noise of the instrument (see table 1.). Note that the low residuals for the field calibration at 150 ppmv are an artefact due to few measurements at this humidity. The average drift observed combining the offset isotopic composition over a month is slightly under 1‰ in $\delta^{18}O$ and reaches 8‰ in δD (type (1) calibration).

_Table 1: Average residuals compared to the quadratic fit toward humidity of laboratory (5 sets) and field calibrations for different humidity levels for the Picarro, cf Figure 4a) and b)._

| | | 200 | 400 | 600 | 800 |
|---|---|---|---|---|---|
| Laboratory calibrations | Humidity (ppmv) | 200 | 400 | 600 | 800 |
| | δD residuals (‰) | 10.1 | 4.9 | 6.0 | 3.1 |
| | $\delta^{18}O$ residuals (‰) | 0.3 | 0.7 | 0.5 | 0.3 |
| Field calibrations | Humidity (ppmv) | 150 | 350 | 480 | 710 |
| | δD residuals (‰) | 1.0 | 6.8 | 2.9 | 5.1 |
| | $\delta^{18}O$ residuals (‰) | 0.6 | 1.0 | 0.5 | 0.4 |

Field calibration could only be performed after the 7[th] of January when the dry air bottle was delivered to Concordia. Then, 2 calibrations per day were realised as follow: 30 minutes calibration, 30 minutes measurements of outside air and 30 minutes calibration. As the data are interpolated on an hourly resolution, this procedure prevents gaps in the data. Altogether, 20 calibrations were achieved from January 7[th] to January 17[th] with two working standards. These logistical issues require adjustment to the calibration procedure described above. Because type (1) calibration could not be performed during the field campaign, we use the drift evaluated from the laboratory calibrations to bracket the maximum drift expected over a period of a month. This results in an important increase of the uncertainty of the measurement of $\delta^{18}O$ from 0.2‰ (optimal value from the Allan Variance) to 1‰ (estimated from the drift of the instrument during the laboratory type (1) calibration) and in δD from 1.3‰ to 6‰.

Type (2) calibration was realised on the field using two working standards calibrated against VSMOW-SLAP: NEEM and UL1 at the end of the campaign. Because the vapour isotopic composition at Dome C was much lower than expected (well below the SLAP isotopic composition), in order to properly estimate the isotope-isotope response of the instrument, it was necessary to evaluate the relevance of the correction obtained from the field calibration. This is described in section 2.6 and required to produce new standards with isotopic composition below the SLAP value. As described in section 2.6, we validated that even by calibrating the isotope-isotope response of the instrument above the SLAP composition; the linearity of the instrument was good enough to extend the calibration down at least to -80‰ in $\delta^{18}O$.

As it was not possible to perform relevant ramps of humidity within one day, type (3) calibration was realised by merging all calibration realised on the field into one series (Figure 4, light colour points). This merged field calibration set provides with an estimate of the linear correction to be applied on the measured humidity (cf supplementary material part 2). The merged field calibration series also documents the non-linearity of the instrument as a function of the background humidity level and is used to correct the values of δD and $\delta^{18}O$ measurements in water vapour. The laboratory and field calibrations do not match. Calibrations realised in the lab and in the field have been reported to differ (Aemisegger et al., 2012) which rules out the use of pre-campaign laboratory calibrations, even though laboratory calibration is still useful to provide insight in the minimum error to be expected during the field campaign. There is no indication from Aemisegger et al. (2012) that opposite trends were obtained during the different calibrations. We checked the possibility that this behaviour could be linked with the remaining water content of the air carrier as it occurred for Bonne et al. (2014) for instance at low humidities. For both field and laboratory calibrations, we used Air Liquide Alphagaz 1 air with a remaining water content below 3ppmv. One possible explanation for the opposite trend on the field compared to laboratory calibrations could be an extraordinary isotopic composition of the air carrier from the dry air cylinder during the field campaign. However, we do not believe the air carrier is responsible for this opposite trend. First, we realised a calculation of the isotopic composition of the 3 ppmv of water remaining in the cylinder necessary to explain the difference between the field and the laboratory calibrations trends. The calculation is the average of the isotopic composition weighted by the water content between the remaining 3 ppmv (unknown isotopic composition to be determined) and the water vapour generated by the calibration device (known humidity and isotopic composition). It is not possible to find one unique value matching the system and the range of calculated values spans between $\delta^{18}O$ = -450‰ and $\delta^{18}O$ = -650‰. This range is beyond anything observed from regular use of air carrier cylinder. Second, the same cylinder was used during another campaign and a similar feature was not observed (not shown). Finally, we observe a very good agreement between the results from the Picarro and the cryogenic trapping data (see

section 2.6 and 3.1) with a difference of 1.16‰ for $\delta^{18}$O using the field calibrations. If we use the laboratory calibrations, this would create a much larger difference (above 5‰ difference in $\delta^{18}$O) which validates the calibration procedure and the use of the field calibration. Here, we attribute this odd behaviour of the isotope-humidity response to the important amount of vibration in the shelter and therefore decided to use this isotope-humidity response to calibrate the dataset. Indeed, this response should be representative of the global behaviour of the Picarro measuring during this campaign.

To summarise, here we cannot estimate from these measurements the drift over the period of field measurement. However, we incorporate an uncertainty for this drift from the laboratory calibrations. These laboratory calibrations were realised on a period longer than the campaign and therefore should bracket the actual drift of our instrument during field deployment and decrease the accuracy of the measurement to 1‰ in $\delta^{18}$O and 8‰ in $\delta$D.

The precision on the absolute value is calculated from the largest residuals of both the laboratory and field calibration fit. It rises up to 18‰ for $\delta$D at 200ppmv and 1.7‰ for $\delta^{18}$O at 400ppmv, with better precision at higher humidity (Figure 4). This highlights the need for regular calibrations to obtain the best performances, unfortunately with a very high cost for this study: the lack of regular calibrations hinders by a factor of 5 the precision of the measurements (1.3‰ for $\delta$D in the best conditions from the Allan Variance against 6‰ for $\delta$D from the mean residuals of the calibration). Additional information about the linearity of Picarro infrared spectrometers against the SMOW-SLAP scales at isotopic composition below the SLAP values can be found in section 2.6 with the description of the measurements of the cryogenic trapping samples.

### 2.5.    Data post-treatment and performances

In addition to the calibration and averaging necessary to improve the accuracy and precision of the dataset, we had to correct our data from the introduction of condensate inside the inlet. Figure 5 illustrates two of such "snow-intake" events, providing typical examples of duration and shape. Indeed, our inlet was facing the dominant wind without any protection to prevent introduction of condensates. Such protection usually requires to be heated to prevent condensation of water vapour under supersaturated conditions; however heating would lead to sublimation of all the precipitation falling into the inlet, which would then increase the vapour content. Moreover, micro droplets or crystals are often floating in the air on the Antarctic Plateau, and reduce the efficiency of any precipitation filter. We therefore decided to remove the effect of all sort of precipitation events through a post-treatment of our datasets. This is justified by a small number of cases (fewer than 100), clearly identified as "snow-intake" events.

A manual post-treatment was thus realised following systematic rules. All data with a specific humidity higher than 1000ppmv were discarded: this value was chosen as the maximum surface air temperature observed during the campaign (-24.6°C) implies a theoretical maximum saturated vapour content of 1030 ppmv. After this first post treatment, the largest humidity measurements of 977 ppmv, slightly lower than the maximum saturated vapour content suggesting that we may have discarded only a few relevant high humidity data in our post-processing.

All humidity peaks higher than natural variability were also discarded, using as a threshold 5 times the standard deviation in normal conditions (which is between 10 and 20ppmv). In very few occasions (only 2 during the entire campaign), a very high density of snowflakes could create a regular inflow of snow in the inlet, leading to an increase of the vapour content without peak shapes. In those cases, the amplitude and the frequency of the specific humidity variability still allowed us to distinguish precipitation introduction from the "background" vapour signal. These periods associated with important "snow-intakes" created gaps in the dataset (4 hours in total). Gaps in our dataset mostly arise from calibration of the instruments and power shortages (30 to 60 minutes gaps) that could be filled by interpolating.

Two running averages were performed: first at 10 minutes resolution, without filling the gaps which corresponds to approximately 3% of the dataset (Figure 5), then an average at a resolution of the hour where the gaps were filled by linear interpolation (only 1% of the whole datasets had gaps superior than an hour) apart from the 13$^{th}$ of January when 4 hours in a raw were missing due to intense precipitation event. Finally, 0.7% of the dataset is missing at the 1 hour resolution.

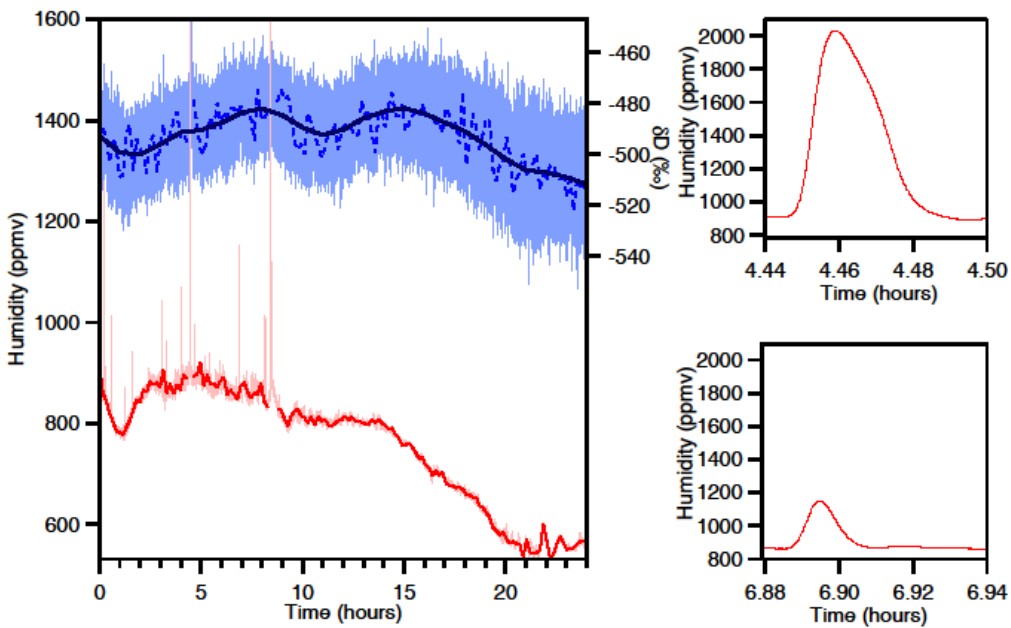

*Figure 5: left : example of raw data measured by the Picarro: humidity (light red, ppmv) and δD (light blue, ‰), data averaged over 10 minutes for humidity (red, ppmv) and for δD (dashed line blue, ‰) and over 1 hour for δD (dark blue, ‰). Right: zoom on two "precipitation events" identified in the humidity signal of the Picarro (top, snow flake; bottom, diamond dust).*

Even though the spectrometer was located at the border of the clean area of the station, we verified that the influence of the station did not contaminate the vapour by analysing wind direction. As mentioned earlier, the shelter is almost 1km upstream the station against the dominant wind. Few events with wind direction pointing from the station were identified (21 hours spread over 5 days during the whole campaign where the wind direction is pointing from the station plus or minus 20°). Most of these events match the period when the wind speed was very low (< 2 m.s$^{-1}$). We used the methane measurements also provided by the Picarro L2130 in parallel with the vapour measurements to assess any potential anthropic contamination of the vapour at the shelter area. An anthropic contamination of the vapour could lead to artificial values of isotopic composition. Indeed, combustion of fossil fuels have been shown to produce d-excess for instance (Gorski et al., 2015). Small spikes of methane were detected for only two occurrences: December the 28$^{th}$ between 9:30 and 10:40 and January the 3$^{rd}$ between 6:00 and 7:00 (local time). They match events with wind direction pointing at the shelter. These two events were fairly short and no specific impact on either humidity or isotopic composition can be identified for these events.

### 2.6. Cryogenic trapping of the moisture

Water vapour was trapped with a cryogenic trapping device (Craig, 1965) consisting of a glass trap immersed in cryogenic ethanol. Cryogenic trapping has been proven reliable to trap all the moisture contained in the air and therefore to store ice samples with the same isotopic composition as the initial vapour (He and Smith, 1999; Schoch-Fischer et al., 1983; Steen-Larsen et al., 2011; Uemura et al., 2008). Two different cryogenic trapping setups have been deployed. The first one in 2006/07 was based on traps without glass balls. These

traps cannot be used with air flow above 6 L/min in order to trap all the moisture because the surface available for thermal transfer is rather small. In order to be certain of trapping all the moisture, two traps in series were installed. Because of the lack of glass balls, the absence of water in the trap at the end of the detrapping can be observed. This was a very important validation because detrapping efficiency is essential to obtain correct values of isotopic composition (Uemura et al., 2008). During the second campaign, we used traps filled with glass balls to increase the surface available for thermal transfer and therefore that can be used at higher flows. This cryogenic trapping setup relies on extensive tests previous to the campaign indicating that our custom-made glass traps filled with glass balls at -100°C successfully condensates all the moisture even for a flow up to 20L/min. These tests have been realised with 1. a Picarro (L2140i) to attest that the remaining humidity was below the measurement limit (around 30 ppmv) and 2. with a second trap downstream to evaluate the presence of ice after a period of 12 hours which would indicate a partial vapour trapping. These tests enable to validate the system we used, similar to Steen-Larsen et al. (2011), and motivate its deployment for the second campaign at Dome C. Extensive tests have also proven that complete detrapping can be done with traps filled with glass balls despite no direct observation of possible remaining water. The results shown later on (Figure 10) shows that similar values are obtained from both types of setup (with or without glass balls) and assess the reliability of both the methods.

Here, we present the results of two cryogenic trapping campaigns: one in 2006/07 and one in 2014/15. During the 2006/2007 campaign, 20 samples were gathered by cold traps (without glass balls) immersed in ethanol at -77°C, with a pump with a flow of 6L/min and 36hour sampling periods. For the campaign of 2014/2015, 20 samples were gathered by cold traps (filled with glass balls) immersed in ethanol at -100°C under a flow of 18L/min and 10 to 14 hours trapping periods. The samples were extracted from the traps by heating them up to 200°C on a line under vacuum connected to a glass phial immersed in the cryogenic ethanol for 10 to 12 hours. This process allows the total transfer of the water by forced diffusion and produces samples between 2 to 4 mL. On January the 8[th] 2015, the high flux pump was damaged and was replaced by a membrane vacuum pump with only 8L/min flow increasing the trapping duration to 24 to 36 hours.

As no particles filter was installed on the inlet (cf section 2.1.), we trapped both the precipitation captured by the inlet and the surface vapour. This might lead to biases when precipitation occurred, and must be taken into account when comparing the results between the spectrometers and the cold trap.

Samples from the 2014/2015 campaign were then shipped for laboratory analyses using a Picarro L2140i. The samples were injected through a syringe in a vaporiser and an auto-sampler. The classical calibration procedure to analyse polar samples is using three internal standards calibrated against SMOW and SLAP (Standard Mean Ocean Water and Standard Low Antarctic Precipitation): NEEM ($\delta^{18}O$ = -33.56‰ and $\delta D$ = -257.6‰), ROSS ($\delta^{18}O$ = -18.75‰ and $\delta D$ = -144.6‰) and OC3 ($\delta^{18}O$ = -54.05‰ and $\delta D$ = -424.1‰). The isotopic composition of the sample to analyse has to be surrounded by the isotopic composition of the standards for the calibration to be efficient. As the isotopic composition of the vapour in Concordia is well below SLAP ($\delta^{18}O$ = -55.50‰ and $\delta D$ = -427.5‰), i.e. $\delta^{18}O$ is around -70‰, no standard was available to bracket the sample isotopic composition. It was therefore important to check the linearity of the instruments for $\delta^{18}O$ values below -55‰.

In order to do so, we prepared new home made standards: we diluted a known home-made standard EPB ($\delta^{18}O$ =-7.54 ± 0.05‰) with highly depleted water Isotec Water-$^{16}O$ from Sigma-Aldrich (99.99% of $^{16}O$ atoms, here after DW for depleted water). We first had to determine the absolute composition of the depleted water by realising several dilutions of the water with isotopic composition in the range between SMOW and SLAP. The dilution was realised with a Sartorius ME215P scale, whose internal precision is certified at 0.02mg. The water was injected through needles in a glass bottles covered by paraffin films to prevent evaporation. All the weights were measured 4 times in order to improve the precision of the measurements. From the different measurements, the accuracy is estimated at 0.1mg after correcting for the weight of the air removed from the bottle by injecting the water. Four new home-made standards were realised in the range SMOW/SLAP and measured 15 times each with a Picarro L2140i (cf Figure 6, part 1). Their isotopic

composition is scattered along the line from the EPB composition to the depleted water composition. Because we know the exact dilution of EPB with the depleted water, we can use the measured $\delta^{18}O$ values to precisely infer the isotopic composition of the depleted water $\delta^{18}O_{DW}$ or $R_{DW}^{18} = (\delta^{18}O_{DW}/1000 + 1) * R_{SMOW}$ where $R_{SMOW}^{18} = 2005.2$ is the absolute isotopic composition of the SMOW in $H_2^{18}O$.

The isotopic composition of the mix is given by:

$$\delta^{18}O_{mix} = \delta^{18}O_{EPB} + \frac{R_{DW}^{18} - R_{EPB}^{18}}{R_{SMOW}^{18}} X_{DW} \qquad (1)$$

where $X_{DW}$ is the ratio of quantities of depleted water vs EPB in the dilution. The slope of the linear regression of $\delta^{18}O_{mix}$ with $X_{DW}$ provides directly an estimate of the isotopic composition of the depleted water. We find $R_{DW}^{18} = 128 \pm 2$ (equivalent to $\delta^{18}O_{DW} = -936.2 \pm 0.6$ ‰), hence slightly less depleted than the specifications given by the producer (purity of 99.99%). Another determination can be done independently by using the formula (1) for one single dilution. Using independent dilutions done within the range SMOW/SLAP, we obtain $R^{18}_{DW}= 127$ and $130$.

In a second step, we produce 3 other water homemade standards by dilution of EPB with "almost pure" $H_2^{16}O$ to obtain $\delta^{18}O$ values below SLAP. Using the know dilution amount and the isotopic ratio of "almost pure" $H_2^{16}O$ determined above, we compare the measurements for these 3 homemade standards, i.e. placed on a SMOW-SLAP scale with classical calibration procedure to the values calculated using formula (1) (Figure 6, part 2.). Given the precision on the isotopic ratio of the "almost pure" $H_2^{16}O$, on the EPB and the precision of the scale, the precision of the calculation of $\delta^{18}O_{mix}$ is 0.05‰ (uncertainty propagation in formula (1)).

Residuals between measured and calculated $\delta^{18}O$ are less than 0.2‰ for the homemade standards at -60‰ and -80‰ and less than 0.3‰ at -110‰. We thus conclude that the Picarro L2140i can be used safely to infer linearly $\delta^{18}O$ values down to -80‰ which encompasses the $\delta^{18}O$ range of our water vapour samples; and is close to be linear for $\delta^{18}O$ values down to -110‰ (deviation of 0.3‰ slightly higher than the measurement uncertainty).

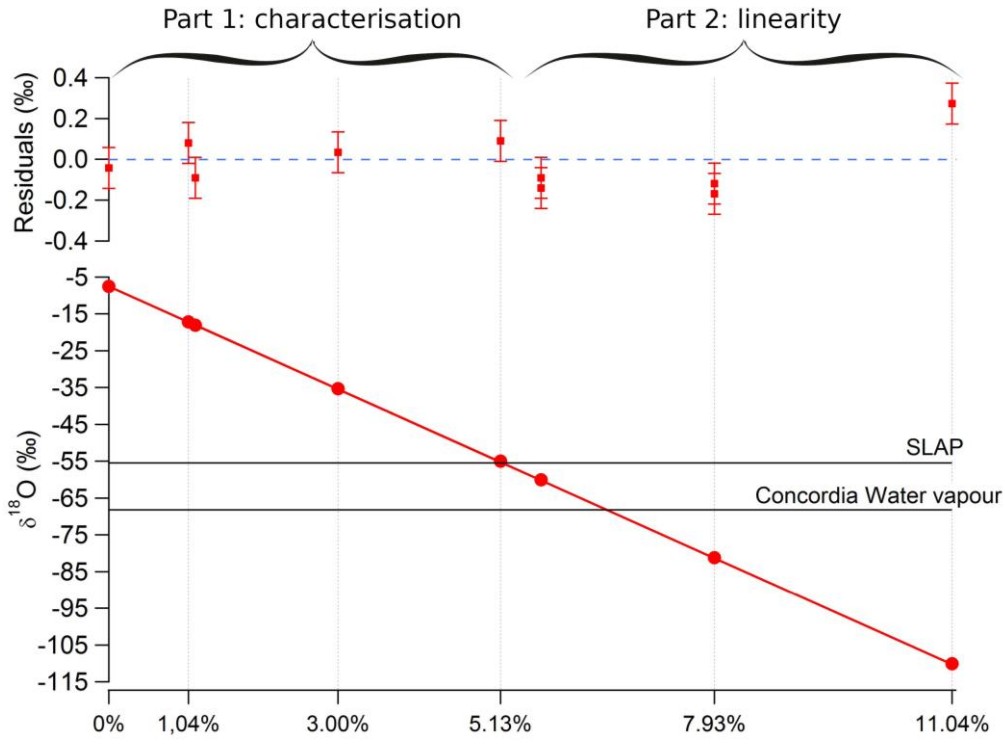

*Figure 6: Isotopic composition measured by liquid injection in the Picarro L2140i for different samples prepared by dilution of EPB with "almost pure" water, the red dots are the measurements, the red line is the*

*calculated isotopic composition and the red squares for residuals are the difference between the measurements and the theoretical composition.*

## 3. Results

### 3.1. Validation of infrared spectrometry data

The data gathered by the cold trap and the infrared spectrometers during the 2014/2015 campaign are displayed in Figure 7.

The measurements performed by the Picarro (light lines) from the 25$^{th}$ of December to the 4$^{th}$ of January are marked by a 10‰ gradual decline in $\delta^{18}O$, and a 40‰ gradual increase in d-excess. By contrast, the second 
part of the measurements (performed after the 4$^{th}$ of January) does not show any long term multi-day trend. We also observe a decrease in $\delta^{18}O$ and an increase in d-excess in the cold trap data from December 25$^{th}$ to January 5$^{th}$. These decrease in $\delta^{18}O$ and increase in d-excess are also recorded in the period from the 5$^{th}$ of January to the 13$^{th}$ of January in the cold trap results while they are not observed in the Picarro data.

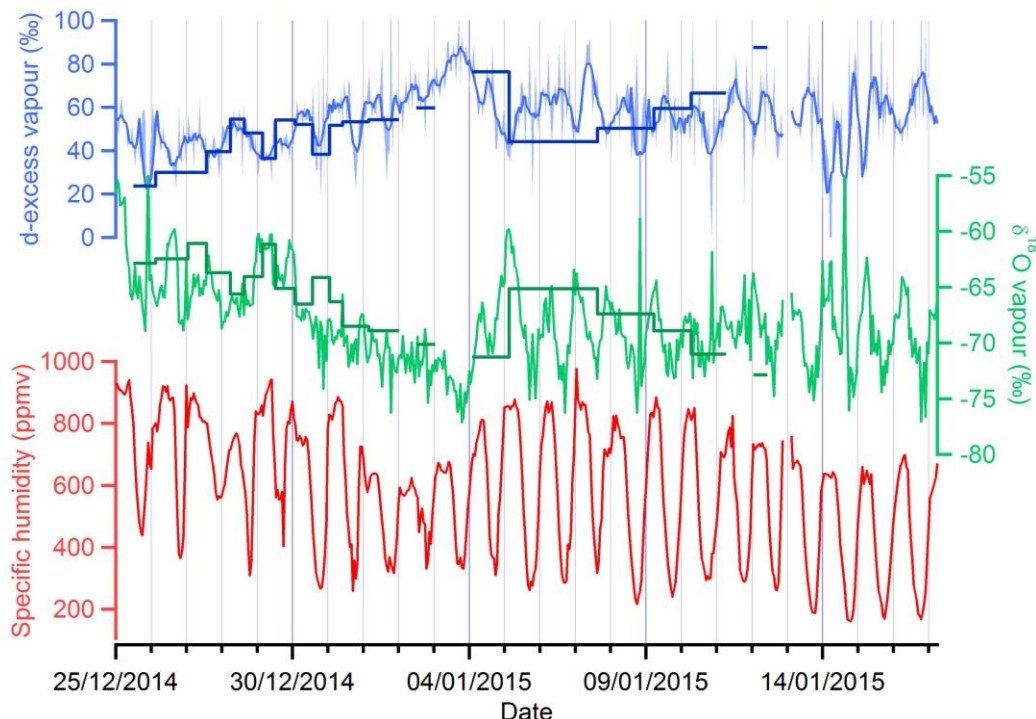

*Figure 7: hourly average $\delta^{18}O$ (‰) in green, raw d-excess (‰) in light blue (d-excess smoothed on a 3 hours span in thick blue), and hourly average of the specific humidity (ppmv) in red during the campaign 2014/2015. Measurements by the Picarro are displayed as the thin light lines and measurements performed in the laboratory from the cold trap samples are displayed as dark bars.*

During a similar campaign from Greenland (Steen-Larsen et al., 2011), differences between infrared spectrometry in situ and cryogenic trapping measurements were generally around 0.1 ‰ in $\delta^{18}O$. In comparison, we observe that the cold trap $\delta^{18}O$ values are generally higher than the $\delta^{18}O$ measured by the 
Picarro. This can be explained by several factors. First, the isotopic composition sampled using the cold trap

is weighted by humidity: the cold trap is trapping more moisture when the humidity is highest, which also corresponds to the moment when the isotopic composition is the highest. In order to take this into account, we weighted the isotopic composition from the Picarro by specific humidity (not shown). In average, the weighted isotopic composition has an offset of +1.1‰ in $\delta^{18}$O compared with the original dataset, rising up to 7.2‰ on December the 31$^{st}$ and down to -2.9‰ on January the 6$^{th}$. In this case, the cold trap $\delta^{18}$O is still in average higher than the isotopic composition weighted by humidity, with an offset of +1.16‰ for $\delta^{18}$O and -3‰ for d-excess, which lies within the error bar of our measurements. We thus conclude that, at first order, our cold trap measurements validate the laser spectrometer data.

The cold trap measurements may also include snow-intake events that were captured by the inlet, whereas we removed such data in the spectrometer measurements. Because the isotopic composition of precipitation is enriched compared to the vapour, the introduction of snow crystals in the cold trap inlet could explain a small part of the positive offset of cold trap measurements compared to the infrared spectrometry. No quantitative estimation of this bias has been realised.

### 3.2. Two climatic regimes

Figure 8 presents the specific humidity and isotopic composition ($\delta^{18}$O, $\delta$D and d-excess) measured by the Picarro. The data are continuous from December 25$^{th}$ 2014 to January 17$^{th}$ 2015, except for 4 hours on January 13$^{th}$ due to a large snowfall event. These data are compared with the 3m temperature and the 3m wind speed (Section 2.1) and also to the surface temperature monitored by infrared sensing. Note that the different temperature measurements are not intercalibrated and may present a limited bias of 1°C. Table 2 summarizes the average, minimum and maximum values for 3m temperature, surface temperature, humidity and isotopic composition.

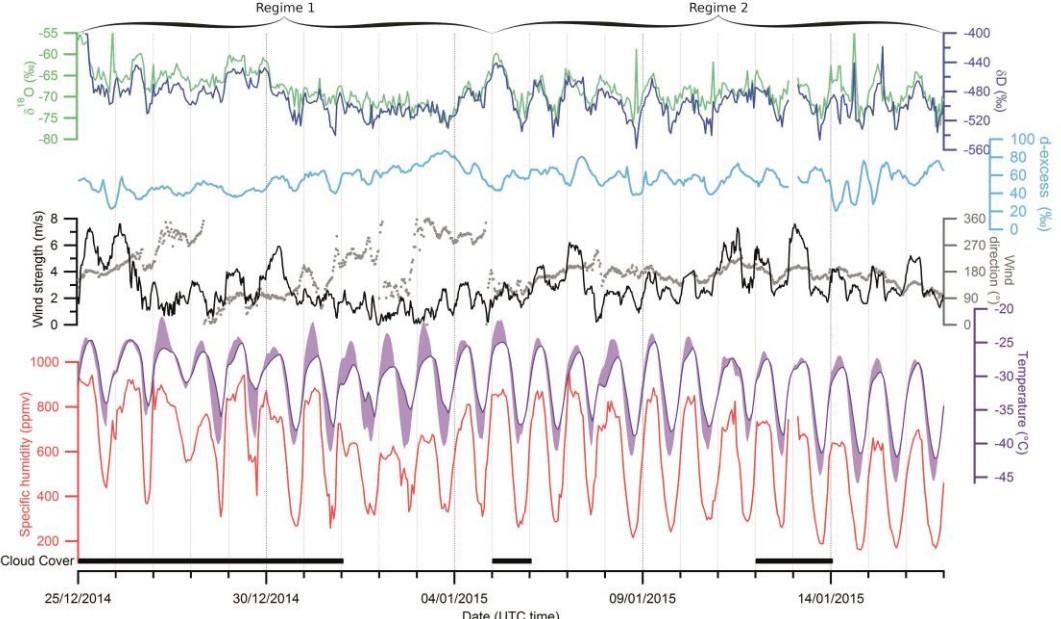

*Figure 8: hourly average $\delta$D (‰) in dark blue, hourly average $\delta^{18}$O (‰) in green, d-excess (‰) smoothed on a 3 hours span in light blue, and hourly average of the specific humidity (ppmv) in red measured by the Picarro during the campaign; comparison with 3m temperature (purple, °C), difference between ground and 3m temperature (purple shade, °C), wind direction (grey dots, °) and speed (black line).*

*Table 2: Average, minimum and maximum values over the whole campaign for air temperature ($T_{3m}$), snow surface temperature ($T_{surf}$), specific humidity (q), $\delta$D (‰), $\delta^{18}$O (‰), and 3 hour smoothed d-excess (‰).*

|  | Average | Minimum | Maximum |
|---|---|---|---|
| $T_{3m}$ (°C) | -31.2 | -42.6 | -24.6 |
| $T_{surf}$ (°C) | -31.5 | -46.1 | -21.2 |
| q (ppmv) | 589 | 161 | 977 |
| $\delta D$ (‰) | -491 | -558 | -393 |
| $\delta^{18}O$ (‰) | -68.2 | -77.1 | -53.9 |
| d-ex (‰) | 55.1 | 21 | 88 |

Even though the sun is never actually passing below the horizon, when the zenithal angle is low, snow surface radiation deficit generates a strong radiative cooling of the surface which leads to stratification of the atmospheric boundary layer. Daily cycles are clearly visible in all the variables. Greater diurnal temperature variations are observed at the surface than at 3 meters even though average temperatures remain similar as already observed in Kohnen (van As et al., 2006). Day temperature at the surface rises up to 8°C higher than at 3 meters during the period from December 26[th] 2014 to January 4[th] 2015. After January the 4[th], differences remain small (less than 2°C). This first difference will lead us to distinguish the two regimes to further investigate: a first one from December 26[th] 2014 to January 4[th] 2015, and a second one from January the 5[th] to January the 17[th] 2015.

Table 3 compares the average values, the diurnal amplitudes and the trends within the different datasets. Temperature is higher during Regime 1, probably due to the proximity to the solar solstice. Diurnal amplitudes in air temperature and humidity are significantly higher in regime 2 than in regime 1. In regime 1, isotopic daily cycles are dumped and completely erased from the 1[st] of January to the 3[rd] of January, whereas daily cycles are important for regime 2 (in phase with those of temperature); a significant day to day trend appears during the regime 1 with almost -1‰/day for $\delta^{18}O$ and is not present in the regime 2 (0.07‰/day for $\delta^{18}O$).

*Table 3: Average, daily amplitude and daily trend over the whole campaign for air temperature ($T_{3m}$, °C), snow surface temperature ($T_{surf}$, °C), specific humidity (q, ppmv), $\delta D$ (‰), $\delta^{18}O$ (‰), and smoothed d-excess (‰).*

|  | Regime 1 : from 26/12 to 04/01 | | | Regime 2 : from 05/01 to 17/01 | | |
|---|---|---|---|---|---|---|
|  | Average | Amplitude | Trend (/day) | Average | Amplitude | Trend (/day) |
| $T_{3m}$ (°C) | -29.9 | 7.6 ± 0.2 | -0.29 ± 0.02 | -32.4 | 11.9 ± 0.2 | -0.38 ± 0.02 |
| $T_{surf}$ (°C) | -30.2 | 14.2 ± 0.4 | -0.34 ± 0.05 | -32.6 | 16.2 ± 0.3 | -0.47 ± 0.03 |
| q (ppmv) | 631 | 341 ± 20 | -24 ± 3 | 541 | 521 ± 13 | - 39 ± 2 |
| $\delta D$ (‰) | -490 | 14 ± 3 | -3.7 ± 0.4 | -495 | 38 ± 2 | -0.8 ± 0.3 |
| $\delta^{18}O$ (‰) | -68.1 | 1.4 ± 0.6 | -0.92 ± 0.06 | -68.9 | 5.4 ± 0.4 | -0.07 ± 0.04 |
| d-ex (‰) | 54.9 | 8 ± 1 | 3.7 ± 0.2 | 56.2 | 13 ± 2 | -0.2 ± 0.2 |

We attribute the difference between the two regimes to changes in atmospheric stability, in particular during the "night". Indeed, during day time, the convection enables strong mixing in both regime 1 and regime 2. However, significant differences are noticeable in the nocturnal stability between regime 1 and 2 which impact the night-time turbulent mixing.

Atmospheric static stability is further assessed using the Richardson number (Richardson, 1920), which is a ratio between the square of the Brunt-Väisälä frequency ($N = \sqrt{\frac{g}{\theta} \frac{d\theta}{dz}}$ where $\theta = T(P_0/P)^{R/c_P}$ is the potential temperature calculated from $P_0$ the standard reference pressure, $R$ the gas constant of air and $c_P$ the specific heat capacity) and the square of the horizontal wind gradient (see supplementary material part 2). During regime 1, the Richardson number experiences important daily cycles, rising higher than 0.2 during night time, indicating a stable and well stratified boundary layer and dropping lower than 0 during daytime

indicating a non-stable, convective atmosphere (King et al., 2006). The Richardson number is in particular really large for the nights from 1[st] of January to the 3[rd] of January (rising up to 0.85) highlighting an enhanced nighttime stratification during this period. The regime 1 is so characterized by a well-marked diurnal cycle with a convective activity during the "day" and a well stably stratified atmospheric boundary layer during the "night". By contrast, the Richardson number is lower during the night in regime 2 which leads to smaller diurnal cycles of stratification. This can be explained by stronger winds during the nights in regime 2 (Figure 9) which enhance the turbulent mixing in the atmospheric boundary layer and tends to reduce the stratification.

We now investigate the mean daily cycle of all data during each regime. For this purpose, the trend is removed by subtracting to all data the average value of the day. We then produce a mean value for each hour of the day over the whole regime. The correlations between the average daily cycles of isotopic composition, 3m temperature, 3m wind speed and surface temperature are given on Table 4. 3m temperature is less strongly correlated with surface temperature during regime 1 compared to regime 2. During regime 2 nighttime, the atmosphere is more turbulent and therefore atmospheric mixing is more efficient. For a more stratified nocturnal atmosphere (regime 1), we expect surface temperature to be less correlated to 3m temperature and also to isotopic composition.

*Table 4: Slope and correlation coefficient between the different data average daily cycle: for each data, the average of the day was removed and a trend-free daily cycle for each regime was produced*

|  | Regime 1: from 26/12 to 04/01 | | Regime 2: from 05/01 to 17/01 | |
| --- | --- | --- | --- | --- |
|  | Slope | $r^2$ | Slope | $r^2$ |
| $\delta D$ (‰) vs $q$ (ppmv) | $0.043 \pm 0.005$ | 0.79 | $0.071 \pm 0.003$ | 0.96 |
| $\delta D$ (‰) vs $T_{3m}$ (°C) | $2.0 \pm 0.2$ | 0.74 | $3.2 \pm 0.2$ | 0.94 |
| $\delta D$ (‰) vs $T_{surf}$ (°C) | $0.95 \pm 0.2$ | 0.58 | $2.3 \pm 0.1$ | 0.95 |
| $\delta D$ (‰) vs $\delta^{18}O$ (‰) | $6.0 \pm 1.3$ | 0.48 | $6.5 \pm 0.6$ | 0.85 |
| $q$ (ppmv) vs $T_{3m}$ (°C) | $45 \pm 2$ | 0.94 | $44 \pm 2$ | 0.96 |
| $q$ (ppmv) vs $T_{surf}$ (°C) | $24 \pm 2$ | 0.89 | $32 \pm 1$ | 0.98 |
| $T_{3m}$ (°C) vs $T_{surf}$ (°C) | $0.49 \pm 0.05$ | 0.80 | $0.69 \pm 0.04$ | 0.92 |

We also observe that the correlation of surface isotopic composition and temperature, and between $\delta^{18}O$ and $\delta D$ is stronger for regime 2 (turbulent nocturnal atmosphere) than for regime 1 (stratified nocturnal atmosphere). An explanation for this correlation could be the temperature influence on the fractionation at the snow-air interface. In the case of regime 2, as the turbulence allows efficient air mass mixing, the isotopic composition at 2 meters is directly related to what is happening at the surface; hence the isotopic composition is strongly correlated with surface temperature. Such a situation was already described in NEEM station in Greenland (Steen-Larsen et al., 2013), where similar temperature and water vapour isotopic composition cycles were observed during 10 days, leading to the conclusion that the snow surface was acting successively as a sink during the night and as a source during the day. They also emitted the hypothesis than the vapour isotopic composition could be at equilibrium with the snow one, at least during part of the day. Exchange with the vapour could also have strong impact on snow metamorphism in Concordia, as observed in NEEM (Steen-Larsen et al., 2014a).

In the case of regime 1, when atmosphere is at least part of the time stratified, the mixing of the first layers of the atmosphere is not efficiently done by turbulence. In these situations happening mostly at night, the ground is cooling faster than the air above it, creating vertical gradients in moisture content of the atmosphere (van As and van den Broeke, 2006).

We now investigate the timing of the average diurnal cycles (Figure 9). By comparing the position of the maximal slope (which enables a more precise determination of dephasing than the maxima), we notice a shift of approximately 2 hours between surface and 3m temperature. Specific humidity average daily cycle is synchronised with 3m temperature in both regimes 1 and 2. For regime 1, no diurnal cycle appears in surface vapour isotopic composition. For regime 2, the daily cycle of surface vapour isotopic composition is

synchronised with surface temperature, and therefore shifted 2 hours earlier than 3m temperature and humidity. This is consistent with the hypothesis of temperature-driven exchanges of molecules between the air and the snow surface in regime 2. This hypothesis will be discussed in more details in part 3.3.

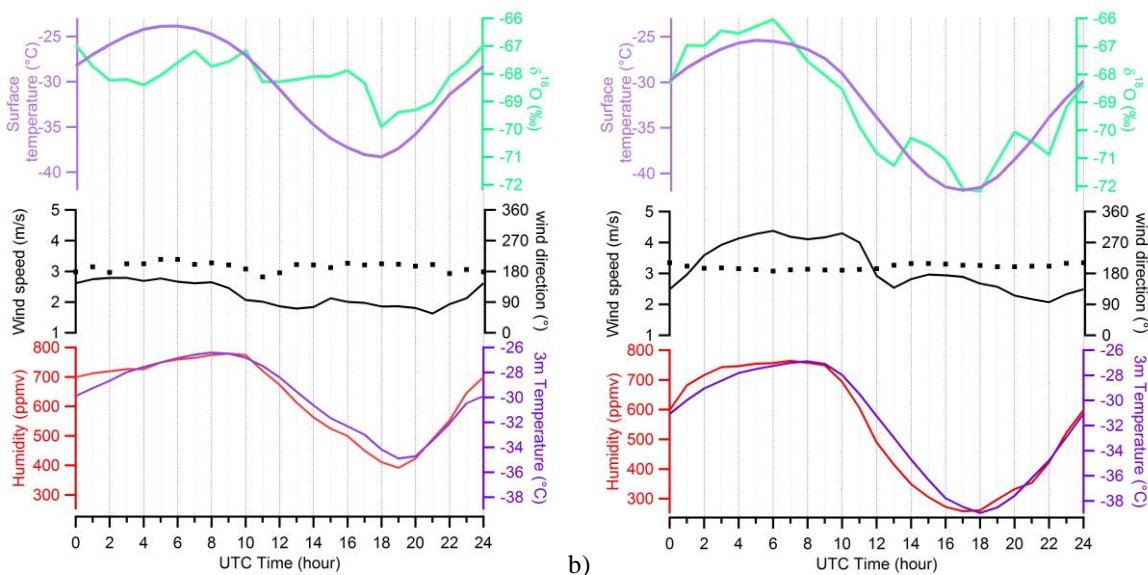

a)   b)

*Figure 9: Comparison of average daily cycles (UTC time) of 3m temperature (light purple, °C), surface temperature (dark purple), specific humidity (red, ppmv), wind speed (black line, m/s), wind direction (black dots, °), $\delta^{18}O$ (green, ‰) for a) regime 1 and b) regime 2.*

The diurnal amplitude that we measured (38‰ for δD in average during the regime 2) is within the range

obtained in previous studies in Greenland. In NEEM, daily cycles up to 36‰ for δD were measured during summer campaigns (Steen-Larsen et al., 2013), much more important than the one on the coastal areas of Greenland with peak to peak amplitudes of variations of 1‰ for $\delta^{18}O$ in Ivittuut, Greenland (Bonne et al., 2014). Similar pattern is observed around Antarctica, near coastal areas, on a ship near Syowa station, isotopic composition variations are dominated by day to day evolution and there are no diurnal cycles (Kurita

et al., under review).

### 3.3.   Local water vapour δD - $\delta^{18}O$ relationship and snow surface interactions

Figure 10 presents the δD and $\delta^{18}O$ isotopic composition during the 2014/2015 campaign, for continuous

measurements, cold trap data, together with earlier cold trap data from 2006/2007. We observe that all these data depict a common range of isotopic composition and align on a similar slope. In this section, we focus on the slope between δD and $\delta^{18}O$ and not on the d-excess. Indeed, the high-values of d-excess are related to the low value of the slope δD versus $\delta^{18}O$ (around 5 compared to the value of 8 used in the d-excess calculation). Note that discussions of d-excess or of the slope between δD and $\delta^{18}O$ are strictly equivalent in this case.

We observe very low (around 5) δD and $\delta^{18}O$ slopes measured using infrared spectroscopy on site and post campaign mass spectrometry of the cryogenic trapping samples (table 5). In fact, publication of the 2006/2007 cold trap data had been postponed until an explanation for such low vapour line was identified, by

fear of sampling vapour from the station generator. As stated in section 2.5, no such contamination occurred. This slope is much lower than observed in Greenland (Bonne et al., 2014; Steen-Larsen et al., 2013). A very low slope for δD vs δ$^{18}$O in water vapour is not unexpected as Dome C is very far on the distillation path and air masses are very depleted in heavy isotopologues (Touzeau et al., 2016). Indeed, for a Rayleigh distillation, the local relative variations of the isotopic composition of δD and δ$^{18}$O are defined by:

$$\frac{d\,\delta D}{d\,\delta^{18}O} = \frac{\alpha_D - 1}{\alpha_{18} - 1}\,\frac{1 + \delta D}{1 + \delta^{18}O} \qquad (2)$$

where $\alpha_D$ and $\alpha_{18}$ are respectively the equilibrium fractionation coefficients of HDO and H$_2$$^{18}$O (Jouzel and Merlivat, 1984). In the average condition of the campaign (T = -31.5°C and isotopic composition from table 2), even if $(\alpha_D - 1)/(\alpha_{18} - 1) = 9.71$, the very low value of δD (around -500‰) brings down the slope δD and δ$^{18}$O to 5.3‰/‰. Note that the important d-excess values obtained in section 3.2. are due to the very low slope between δD and δ$^{18}$O, and not necessarily to important kinetic effects in this case.

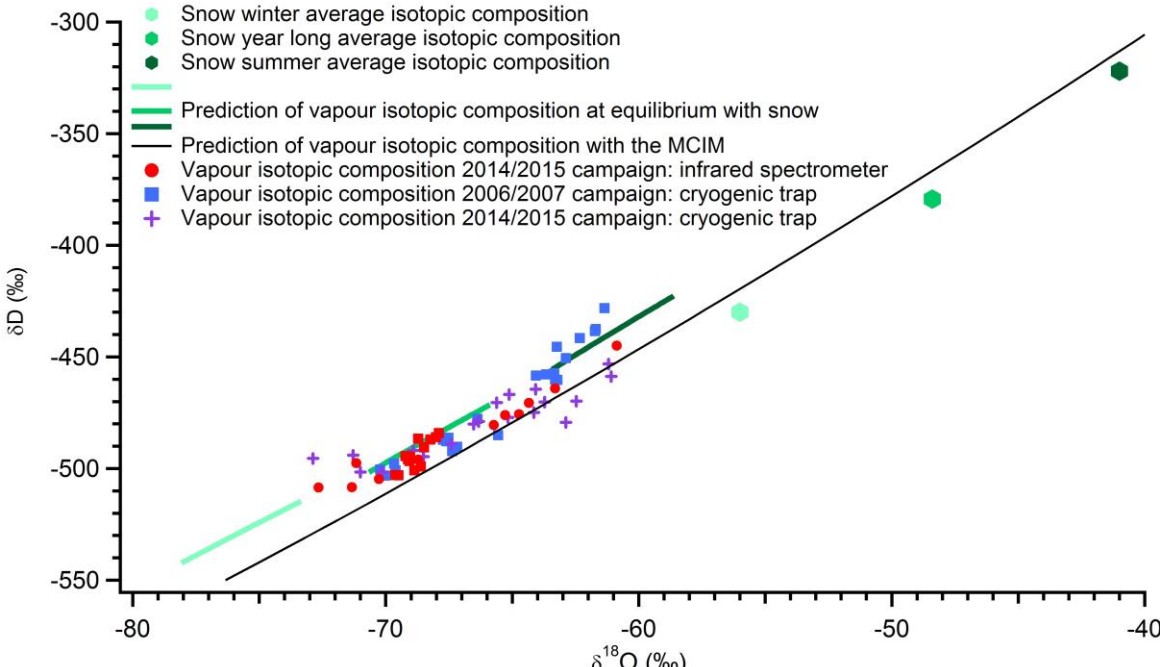

*Figure 10: δD and δ$^{18}$O plots: red is the daily average isotopic composition from the Picarro (circles : regime 1, squares regime 2), purple crosses are the cold trap isotopic composition from 2014/2015 campaign, blue squares are the cold trap isotopic composition from 2006/2007, green hexagons are the isotopic composition of the snow (Touzeau et al., 2015) (light is the average composition minus 1 standard deviation, mid is the average composition and dark is the average composition plus 1 standard deviation), green lines are the respecting calculated equilibrium fractionation in the range of temperature observed during the campaign (Majoube, 1971) (local origin thereafter) and black line is the curve established with a Rayleigh distillation in the MCIM (remote origin thereafter)*

We now discuss in details the possible drivers of the isotopic composition of water vapour at Dome C following several hypotheses: 1-local origin (equilibrium between surface snow and water vapour); 2- remote origin (distillation of a water mass from the coast).

For the first hypothesis, we used the range of annual isotopic composition of the snow at Dome C (Touzeau et al., 2016), represented by green hexagons (average value plus and minus one standard deviation). The slope between δD and δ$^{18}$O of the snow annual isotopic composition is 7.2‰/‰, already lower than 8. From these values, we calculate the corresponding vapour isotopic composition in the range of summer temperature (-20 to -45°C) using standard equilibrium fractionation coefficients (Majoube, 1971; Merlivat

and Nief, 1967). The range of calculated vapour isotopic contents is consistent with observed vapour: from the average value of snow $\delta^{18}O$ = -48.4‰, we get a vapour predicted $\delta^{18}O$ = -68.2‰ at -35°C which lies within the values measured by the Picarro (on average over the campaign $\delta^{18}O$ = -68.9‰). The slope between $\delta D$ and $\delta^{18}O$ on the other hand is higher than the one observed: 6.5‰/‰ vs 5.3‰/‰ for the Picarro and even 4.8‰/‰ for the cold traps. The same calculation with the equilibrium fractionation coefficients from Ellehøj et al. (2013) can predict relevant $\delta^{18}O$ and $\delta D$ values and more realistic slopes (5.7‰/‰).

*Table 5: Slope and correlation coefficients between the different datasets. Picarro and meteorological data are daily average data. Equilibrium fractionation slopes are calculated from the average values (average, plus and minus one standard deviation) with Majoube fractionation coefficients (high M, med M, low M) or Ellehøj fractionation coefficients (med E)*

|  |  | Data for all season | |
|---|---|---|---|
|  |  | **Slope** | **r²** |
| **Picarro data** | $\delta D$ (‰) vs q (ppmv) | $0.12 \pm 0.02$ | 0.61 |
|  | $\delta D$ (‰) vs $T_{3m}$ (°C) | $3.7 \pm 1.5$ | 0.22 |
|  | $\delta D$ (‰) vs $T_{surf}$ (°C) | $4.3 \pm 1.2$ | 0.30 |
|  | $\delta D$ (‰) vs $\delta^{18}O$ (‰) | $5.3 \pm 0.3$ | 0.92 |
|  | q (ppmv) vs $T_{3m}$ (°C) | $43 \pm 6$ | 0.69 |
|  | q (ppmv) vs $T_{surf}$ (°C) | $45 \pm 5$ | 0.79 |
| **Meteo data** | $T_{3m}$ (°C) vs $T_{surf}$ (°C) | $0.7 \pm 0.1$ | 0.63 |
| **Trapping 2006/07** | $\delta D$ (‰) vs $\delta^{18}O$ (‰) | $4.6 \pm 0.7$ | 0.82 |
| **Trapping 2014/15** | $\delta D$ (‰) vs $\delta^{18}O$ (‰) | $4.8 \pm 0.4$ | 0.90 |
| **Equilibrium fractionation** | $\delta D$ (‰) vs $\delta^{18}O$ (‰) high M | 7.02 | Th. |
|  | $\delta D$ (‰) vs $\delta^{18}O$ (‰) med M | 6.50 | Th. |
|  | $\delta D$ (‰) vs $\delta^{18}O$ (‰) low M | 5.99 | Th. |
|  | $\delta D$ (‰) vs $\delta^{18}O$ (‰) med E | 5.65 | Th. |
| **MCIM** | $\delta D$ (‰) vs $\delta^{18}O$ (‰) at -35°C | 6.11 | Th. |

We now analyse the effect of the distillation on the isotopic composition of the water vapour. For this test, we used Mixed Cloud Isotopic Model (MCIM) to compute the isotopic composition of the vapour. The MCIM is a Rayleigh model taking into account microphysical properties of clouds and in particular accounting for mixed phases (Ciais and Jouzel, 1994). The model was tuned with snow isotopic composition of an Antarctic transect from Terra Nova Bay to Dome C to accurately reproduce the isotopic composition of the Antarctic Plateau (Winkler et al., 2012). For instance, the model predicts an average value of snow isotopic composition at Dome C of -51‰ for an average site temperature of -54.5°C when the measurements indicated -50.7‰; note that the model take into account an inversion temperature and that the condensation temperature $T_{cond}$ is deduced from the surface temperature $T_{surf}$ through (Ekaykin and Lipenkov, 2009):

$$T_{cond} = 0.67 \times T_{surf} - 1.2 \qquad (3)$$

The prediction of average vapour isotopic composition by the MCIM is $\delta^{18}O$ = -51.6‰ at -35°C which is much higher than the average vapour measurements ($\delta^{18}O$ = -68.9‰). However, the MCIM manages to predict the isotopic composition of the summer precipitation ($\delta^{18}O$ = -37‰ at -35°C for the model compared to values rising up to -39‰ for matching temperature in Dome C summer precipitation). Therefore, we conclude that the vapour isotopic composition seems to be principally influenced by local effects. Note that the slope between $\delta D$ and $\delta^{18}O$ predicted by the MCIM is around 6.1‰/‰ which is also higher than the one observed during the campaign (between 4.6 and 5.3 for the different datasets).

The precipitation amount in Dome C is less than 10 cm per year (Genthon et al., 2015). Each precipitation event does not form a complete layer of snow and is mixed with earlier snowfall possibly deposited under the earlier winter conditions. The snow isotopic composition is therefore a mix new snowfall and older snow. This phenomenon is amplified by drift and blowing snow (Libois et al., 2014). A mixing between a large range of source isotopic compositions should be considered to compute the local origin hypotheses which
could explain the bias of the slope predicted by equilibrium from a single snow composition compared with experimental data.


## 4. Conclusion

In this study, we assessed the relevance of infrared spectrometry to measure isotopic composition of water at
concentration as low as the ones encountered over the Antarctic Plateau. Apart from the logistic challenges involved in the installation of spectrometers in remote areas; humidity levels, very depleted samples and important local variability imply a technical challenge that the new infrared spectroscopy techniques overcame.

Allan variance measurements in the laboratory indicated the possibility to use Picarro and HiFI spectrometers at humidity as low as 200 ppmv and with almost no loss of precision from 500 ppmv (limit of precision of 0.1‰ $\delta^{18}$O and for 1.1‰ for $\delta$D). Identical measurements in the field showed it was possible to reach similar results in the field even though great care in the environment where the instruments are deployed should be addressed.


For such humidities, the linearity of the instruments is not guaranteed toward humidity and regular calibrations on the field are necessary. In this particular study, it was not possible to calibrate regularly the instruments in the field for logistical reasons, so we bracketed the drift of the instrument by series of calibration in the lab. This is not the optimal method and results in important error bars compared to the
performances of the instrument. The uncertainty of the isotopic composition measurement is therefore 6‰ for $\delta$D and 1‰ for $\delta^{18}$O. We have further validated these measurements through i) a comparison of the data acquired by infrared spectrometry with cryogenic trapping samples, ii) a protocol to calibrate on the SMOW-SLAP scale at $\delta^{18}$O lower than the SLAP $\delta^{18}$O value (-55.5‰). This calibration demonstrated that our Picarro instrument is linear in $\delta^{18}$O down to -80‰ in $\delta^{18}$O and stays almost linear down to -110‰. This is essential
for our study since the mean $\delta^{18}$O value was -68.2‰ at Concordia between December the 25th 2014 and January the 17th 2015.

Two different regimes have been identified during the campaign: a first one from December the 26th 2014 to January the 4th 2015 and a second from January the 5th to January the 17th 2015. The main difference between
the two regimes on isotopic composition is the amplitude of the daily cycles: large and regular during regime 2 and small and irregular in regime 1 and even an almost erased one from January the 1st to January the 4th 2015. For regime 1, correlation of humidity with surface temperature is lowered and isotopic composition is almost stable whereas for regime 2, there is an almost perfect correlation for both humidity and isotopic composition with temperature. We attribute these differences to difference in the stability of the atmosphere.
For regime 1, we explain the drop of correlation by a weakly turbulent boundary layer during which temperature, humidity and isotopic composition diurnal cycles are truncated compared to a second one characterised by efficient turbulence with important diurnal cycles and almost perfect correlation between the snow surface temperature and the first meters of the atmosphere. The second regime therefore appears to be characterised by equilibrium between the isotopic composition of vapour over the first meters and the one of
the snow as already shown for Greenland (Steen-Larsen et al., 2013).

Temperature cycles seem to be directly responsible for isotopic composition cycles, at least in Regime 2, through equilibrium fractionation in sublimation/condensation cycles. At first order, it seems the snow isotopic composition is therefore influencing directly the vapour through fractionation at phase change. The vapour isotopic composition average value matches the one obtained by equilibrium fractionation of the local snow. Still the measured slope between $\delta D$ and $\delta^{18}O$ cannot be explained purely by equilibrium fractionation from local snow. We cannot rule out a contribution of horizontal air advection from inland locations, transported by southward winds and providing small amounts of very depleted moisture.

Finally, our study opens new perspectives in the influence of post deposition effects and their importance for the water stable isotope signal recorded in deep ice cores. In particular, we have shown that the relationship between water vapour $\delta^{18}O$ and temperature can be erased by weakly turbulent regimes. Yearlong monitoring of the isotopic composition of the water vapour could help identify how often these conditions happen, and also if the snow isotopic composition could present biased relationship toward seasonality, temperature or precipitation.

**Author contribution:** M.C., A.L., F.P., S.K. and P.C. prepared the field campaign; M.C. deployed the instruments on the field; VM-D., E.K. and S.K. provided the infrared spectrometers; C.G., L.A., G.P., O.C. and E.V. provided data; M.C. prepared the manuscript with contributions from all co-authors.

**Acknowledgments:** The research leading to these results has received funding from the European Research Council under the European Union's Seventh Framework Programme (FP7/2007-2013) / ERC grant agreement n° [306045]. We acknowledge the programs NIVO and GLACIO and all the IPEV that made this campaign possible, LGGE and LIPHY for providing logistic advice and support. We thank Catherine Ritz, Anais Orsi and Xavier Fain for their help during the preparation of the mission. Many thanks to Philippe Ricaud, Doris Thuillier, Nicolas Caillon, Bruno Jourdain, Olivier Magand and all the eleventh winter-over team for your support and your presence in Concordia. Thanks to Hubert Gallée for all the discussions about polar meteorology.

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
