# Peer review of "Continuous measurements of isotopic composition of water vapour on the East Antarctic Plateau"

_Atmospheric Chemistry and Physics, 2016_

## Referee Comment (RC1) · Anonymous Referee #2 · 21 Apr 2016

This paper presents a highly valuable data set of stable water isotopes measured on the inland Antarctic site where deep ice cores have been retrieved in the past. The measurement captured different meteorological conditions (regime 1 and regime 2) and I'm convinced that there are things we can learn from this data. In particular, the fact that diurnal cycle of water vapor isotopes taken from 2m above the surface corresponds to that of surface temperature rather than air temperature at 3 m above the surface is very interesting for me and may contribute to improve interpretation of ice core records. Furthermore, demonstration of the performance of a commercially available instrument (Picarro L2130-i) at the coldest mountainous region in the world is really valuable for our community. I have no doubt that the authors have done great

work. However, in its current form, it is not easy to read for readers who are not familiar with water vapor isotope measurement using laser instruments. In order to understand their calibration procedure, I have re-read "section 2.4" several times. I strongly suggest that the authors should explain their calibration procedure both in the laboratory and in the field more clearly. The first data becomes benchmarks for the following studies. So, careful explanations are required to show us the robustness of their obtained new data. And, the manuscript seems "methods-heavy" which makes the results and discussion seem a little thin in this current version. Since this is the first data observed on the inland Antarctica, I recommend to add the general description as for observed isotopic features at there. I think the high d-excess values exceeding 50 is noteworthy, if these are true. I therefore recommend publication after addressing the following major points, and a few minor issures.

Major comments

1) Humidity-dependent bias correction The biggest concern of this study is a robustness of correction functions for humidity effects. As far as I know, the shape of this function (gradually increasing trend of isotopic values with decreasing humidity) is common for Picarro L2130-i (see Aemisegger et al., 2012, Bastrikov et al., 2014, Steen-Larsen et al., 2014 and so on). However, in Figure 4, the correction function is largely different between the field observation and the pre-campaign laboratory experiment. Surprisingly, the trend of this function for delta-18O obtained from field campaign is opposite. I was looking forward to listening this reason, however there was no constructive explanation in the text. It just noted that "it is not unexpected" with referring to Aemisegger et al., (2012). I really disappointed this reply. Although the correction function can vary in time, as mentioned in Aemisegger et al., (2012), amplitude and shape of the correction function (in particular for latest Picarro) don't change so much. In addition, she pointed out that the effect of remaining water vapor concentration in the carrier gas might be a major source for their observed discrepancy in the correction function (lab vs. field experiment). In Figure 4, both delta-D and delta-18O values measured

in the field are plotted lower than those for laboratory experiment. Furthermore, the isotopic differences between laboratory experiment and field campaign become lager and larger with decreasing humidity. These raise me a concern about the influence of background water vapor with extremely depleted isotopic values in the dry air bottle (carrier gas). Since cylinder air is not completely dry, the influence of background water vapor should be carefully considered for their measurement. I strongly recommend to add the discussion as for the influence of remaining water vapor in the carrier gas (not only field campaign but also laboratory experiment). If they don't know the exact water vapor concentration in the carrier gas, the authors should consider the usage of the correction function obtained from the laboratory experiments as a substitute of that observed in the field.

2) Cryogenic moisture trapping There is a long history for water vapor trapping for isotope analysis. Even in the temperate region, a custom manufactured trap system has been used to satisfy required trapping efficiency, (e.g., Schch-Fischer et al., 1984, He and Smith, 1999, Uemura et al., 2008). Uemura et al. (2008) clearly mentioned that specially designed glass trap and careful treatment is necessary to get precise and accurate data for d-excess investigation. Therefore, the authors also have used custom manufactured trap system and have done some laboratory tests before going to the field. Addition of these informartion in the text must improve or strengthen the reliability of your data. Because vapor trapping system shown in Helliker et al. (2002) did not use in this study, it is not appropriate to refer to his paper here. I'm not sure if the authors used the same system shown in Steen-Larsen et al (2011), but this study also does not support reliability of your data because the temperature at Concordia station is much colder than at NEEM camp and flow rate used in this study is faster than their study.

3) Calibration procedure for laser measurement Because the authors used custom manufactured calibration system, complete description for their calibration procedure is required. However, in its current form, it is very hard to understand their procedure

except for expert of this field. For example, it was very hard to understand the following sentence: "Line 298: a series of calibration was performed in the laboratory from 100 to 1000 ppmv". Without the explanation of "a series of calibration", readers can't understand the meaning of this sentence. In section 2.4, there are several sentences similar to this. I recommend to reorganize and rewrite section 2.4 with the following information. 1. Schematic figure of self-designed calibration devise 2. Outline of the calibration procedure (order of three type of calibrations and their frequency) 3. Operating procedure for each of calibration (don't forget the description for the SMOW-SLAP linearity correction) 4. Isotopic values of standard water for vapor sources 5. Accuracy and precision of this method (Laboratory experiment) 6. Calibration procedure for field campaign 7. Data quality of field data (include uncertainty of d-excess)

Minor concerns

P4, l167: figure 2 -> Figure 2

P6, l213: This noise -> These

P6 l221: realized -> carried out?

P6 l223: realized -> continued?

P6 l237: figure 3 -> Figure 3

P7 l258-259: "Because we are working.." is the same meaning of the following sentence "L261-262: Standard calibration..." Please remove one of the other.

P7 l265: Harvard Apparatus -> add parts number of this item

P8 l268: Bronkhorst devices -> add parts number of it

P12 l440: figure 6 -> Figure 6

P12 l459: figure 6 -> Figure 6

P13 l481: I think date expression is as follows: December 25th; 25th December; 25

December; December the twenty-fifth; the twenty-fifth of December

I feel strange of the following expression: "December the 25th". Please check the expression of date.

P16 l580: figure 9 -> Figure 9

P17 l619: I can't catch the meaning of "slope at the mid height". Please rephrase it.

P16,l621: Figure 9 shows the negative peak of delta-18O at 18:00, corresponding to the minimum of surface temperature. Isn't this peak significant?

P18, Section 3.3: I can't understand why the authors stick to the discussion of slope value for the delta-D-deltaO plot. The slope is sensitive to the systematic bias so that the authors can't escape from the affect of large uncertainty of the measured isotopic values. As shown in Figure 10, I think that the most remarkable features of new data is extremely high d-excess values. So, I recommend to discuss the reason of these high values in here.

References

Aemisegger, F., Sturm, P., Graf, P., Sodemann, H., Pfahl, S., Knohl, A., & Wernli, H. (2012). Measuring variations of $\delta$18O and $\delta$2H in atmospheric water vapour using two commercial laser-based spectrometers: an instrument characterisation study. Atmospheric Measurement Techniques, 5(7), 1491–1511.

Bastrikov, V., Steen-Larsen, H. C., Masson-Delmotte, V., Gribanov, K., Cattani, O., Jouzel, J., & Zakharov, V. (2014). Continuous measurements of atmospheric water vapour isotopes in western Siberia (Kourovka). Atmospheric Measurement Techniques, 7(6), 1763–1776.

He, H., & Smith, R. B. (1999). Stable isotope composition of water vapor in the atmospheric boundary layer above the forests of New England. Journal of Geophysical Research: Atmospheres, 104(D), 11657.

Helliker, B. R., Roden, J. S., Cook, C., & Ehleringer, J. R. (2002). A rapid and precise method for sampling and determining the oxygen isotope ratio of atmospheric water vapor. Rapid Communications in Mass Spectrometry, 16(10), 929–932.

Schoch-Fischer, H., K. Rozanski, H. Jacob, C. Sonntag, J. Jouzel, G. Ostlund, and M. Geyh. (1984). Hydrometeorologicalfactorscon- trolling the time variation of D,18 O and 3 H in atmospheric water vapour and precipitation in the northern westwind belt. Isotope Hydrology 1983, IAEA-publication, Vienna, Austria, 3–30

Steen-Larsen, H. C., Masson-Delmotte, V., Sjolte, J., Johnsen, S. J., Vinther, B. M., Breon, F. M., et al. (2011). Understanding the climatic signal in the water stable isotope records from the NEEM shallow firn/ice cores in northwest Greenland. Journal of Geophysical Research, 116(D6), D06108.

Steen-Larsen, H. C., Sveinbjörnsdóttir, A. E., Peters, A. J., Masson-Delmotte, V., Guishard, M. P., Hsiao, G., et al. (2014). Climatic controls on water vapor deuterium excess in the marine boundary layer of the North Atlantic based on 500 days of in situ, continuous measurements. Atmospheric Chemistry and Physics, 14(15), 7741–7756.

Uemura, R., Matsui, Y., Yoshimura, K., Motoyama, H., & Yoshida, N. (2008). Evidence of deuterium excess in water vapor as an indicator of ocean surface conditions. Journal of Geophysical Research, 113(D19), D19114.

---

## Referee Comment (RC2) · Anonymous Referee #1 · 11 May 2016

To the authors and editor.

I have read the revised manuscript thoroughly, and find the previous reviewers' comments to have been adequately addressed. I recommend publication "as is". The one change i would suggest is that the figures (at least in my version) are somewhat difficult to read – they are fuzzy. Perhaps a different way of exporting them (.eps format?) would help.

---

## Author Comment (AC1) · 24 May 2016

Dear editor,

Please find below our response to the different comments of the anonymous referees. We are grateful for the time they spent reviewing the paper and think the manuscript has been greatly improved thanks to their comment.

Note that a PDF version has been submitted in annex including colours to highlight the answer to the reviewers and figures in order to be more readable.

On the behalf of all the coauthors,

[Figure]

Mathieu Casado

"Anonymous Referee #1

To the authors and editor. I have read the revised manuscript thoroughly, and find the previous reviewers' com- ments to have been adequately addressed. I recommend publication "as is". The one change i would suggest is that the figures (at least in my version) are somewhat diffi- cult to read – they are fuzzy. Perhaps a different way of exporting them (.eps format?) would help"

The authors are very happy the first anonymous reviewer find our manuscript interesting. Indeed, the PDF has very fuzzy version which differ from the original version we had before submission. We will be careful that this issue is corrected after re-submission.
* * *
"Anonymous Referee #2

This paper presents a highly valuable data set of stable water isotopes measured on the inland Antarctic site where deep ice cores have been retrieved in the past. The measurement captured different meteorological conditions (regime 1 and regime 2) and I'm convinced that there are things we can learn from this data. In particular, the fact that diurnal cycle of water vapor isotopes taken from 2m above the surface corresponds to that of surface temperature rather than air temperature at 3 m above the surface is very interesting for me and may contribute to improve interpretation of ice core records. Furthermore, demonstration of the performance of a commercially available instrument (Picarro L2130-i) at the coldest mountainous region in the world is really valuable for our community. I have no doubt that the authors have done great work."

The authors are very happy the second anonymous reviewer find our manuscript interesting and we will try to address the following comments to improve the quality of the

manuscript as he suggested.

———

"However, in its current form, it is not easy to read for readers who are not familiar with water vapor isotope measurement using laser instruments. In order to understand their calibration procedure, I have re-read "section 2.4" several times. I strongly suggest that the authors should explain their calibration procedure both in the laboratory and in the field more clearly. The first data becomes benchmarks for the following studies. So, careful explanations are required to show us the robustness of their obtained new data."

Rewriting the calibration section seems to be necessary indeed, see point 2 of the major comments.

———

"And, the manuscript seems "methods-heavy" which makes the results and discussion seem a little thin in this current version. Since this is the first data observed on the inland Antarctica, I recommend to add the general description as for observed isotopic features at there. I think the high d-excess values exceeding 50 is noteworthy, if these are true. I therefore recommend publication after addressing the following major points, and a few minor issures. Major comments 1) Humidity-dependent bias correction The biggest concern of this study is a robustness of correction functions for humidity effects. As far as I know, the shape of this function (gradually increasing trend of isotopic values with decreasing humidity) is common for Picarro L2130-i (see Aemisegger et al., 2012, Bastrikov et al., 2014, Steen-Larsen et al., 2014 and so on). However, in Figure 4, the correction function is largely different between the field observation and the pre-campaign laboratory experiment. Surprisingly, the trend of this function for delta-18O obtained from field campaign is opposite. I was looking forward to listening this reason, however there was no constructive explanation in the text. It just noted that "it is not unexpected" with referring to Aemisegger et al., (2012). I really disappointed this reply.

Although the correction function can vary in time, as mentioned in Aemisegger et al., (2012), amplitude and shape of the correction function (in particular for latest Picarro) don't change so much. In addition, she pointed out that the effect of remaining water vapor concentration in the carrier gas might be a major source for their observed discrepancy in the correction function (lab vs. field experiment). In Figure 4, both delta-D and delta-18O values measured in the field are plotted lower than those for laboratory experiment. Furthermore, the isotopic differences between laboratory experiment and field campaign become lager and larger with decreasing humidity. These raise me a concern about the influence of background water vapor with extremely depleted isotopic values in the dry air bottle (carrier gas). Since cylinder air is not completely dry, the influence of background water vapor should be carefully considered for their measurement. I strongly recommend to add the discussion as for the influence of remaining water vapor in the carrier gas (not only field campaign but also laboratory experiment). If they don't know the exact water vapor concentration in the carrier gas, the authors should consider the usage of the correction function obtained from the laboratory experiments as a substitute of that observed in the field."

This is a very interesting point. Here, we used a B50 alphagaz 1 air with a remaining water bellow 3ppmv. In order to estimate the impact of this remaining water vapour, we implement a calculation taking into account that what the Picarro is measuring is the dilution between 3 ppmv of dry air carrier and the humidity of moisture generated by the calibration device. We use the laboratory calibration curve to generate reference level for the different humidities and calculate the impact of a dilution with 3ppmv of water vapour in order to test what value of water vapour isotopic composition would be needed for the dry air carrier to generate such a profile. Here, to obtain the values we obtained on the field, we cannot find one isotopic composition of the remaining water and have a large span of values between $\delta$18O = -450‰ and $\delta$18O = -650‰. These estimates are obtained using the average difference between the laboratory values and the field values and it is not possible to find a clear answer within the different values we test (from 150ppmv to 1000ppmv).

Here are the modifications on the main text (line 343): "The laboratory and field calibrations do not match. Calibrations realised in the lab and in the field have been reported to differ (Aemisegger et al., 2012) which rules out the use of pre-campaign laboratory calibrations, even though laboratory calibration is still useful to provide insight in the minimum error to be expected during the field campaign. Still, in the case of Aemisegger et al. (2012), it was never reported to obtained opposite trend during different calibrations. We verify if this behaviour could be linked with the remaining water content of the air carrier as it occurred for Bonne et al. (2014) for instance at low humidities. For both field and laboratory calibrations, we used Air Liquid Alphagaz 1 air with remaining water content below 3ppmv. One possible explanation for the opposite trend on the field compared to laboratory calibrations could be an extraordinary isotopic composition of the air carrier from the dry air cylinder during the field campaign. However, we do not believe the air carrier is responsible for this opposite trend. First, we realised a calculation of the isotopic composition of the 3 ppmv of water remaining in the cylinder necessary to explain the difference between the field and the laboratory calibrations trends. The calculation is the average of the isotopic composition weighted by the water content between the remaining 3 ppmv (unknown isotopic composition to be determined) and the water vapour generated by the calibration device (known humidity and isotopic composition). It is not possible to find one unique value matching the system and the range of calculated values spans between $\delta$18O = -450‰ and $\delta$18O = -650‰. This range is beyond anything observed from regular use of air carrier cylinder. Second, the same cylinder was used during another campaign and a similar feature was not observed (not shown). Finally, we observe a very good agreement between the results from the Picarro and the cryogenic trapping data (see section 2.6 and 3.1) with a difference of 1.16‰ for $\delta$18O using the field calibrations. If we use the laboratory calibrations, this would create a much larger difference (above 5‰ difference in $\delta$18O) which validates the calibration procedure and the use of the field calibration. Here, we attribute this odd behaviour of the isotope-humidity response to the important amount of vibration in the shelter and therefore decided to use this isotope-humidity

response to calibrate the dataset. Indeed, this response should be representative of the global behaviour of the Picarro measuring during this campaign. "

———

"2) Cryogenic moisture trapping There is a long history for water vapor trapping for isotope analysis. Even in the temperate region, a custom manufactured trap system has been used to satisfy required trapping efficiency, (e.g., Schch-Fischer et al., 1984, He and Smith, 1999, Uemura et al., 2008). Uemura et al. (2008) clearly mentioned that specially designed glass trap and careful treatment is necessary to get precise and accurate data for d-excess investigation. Therefore, the authors also have used custom manufactured trap system and have done some laboratory tests before going to the field. Addition of these informartion in the text must improve or strengthen the reliability of your data. Because vapor trapping system shown in Helliker et al. (2002) did not use in this study, it is not appropriate to refer to his paper here. I'm not sure if the authors used the same system shown in Steen-Larsen et al (2011), but this study also does not support reliability of your data because the temperature at Concordia station is much colder than at NEEM camp and flow rate used in this study is faster than their study."

This is a very good point and has been mentioned in the manuscript (line 433) : "Water vapour was trapped with a cryogenic trapping device (Craig, 1965) consisting of a glass trap immersed in cryogenic ethanol. Cryogenic trapping has been proven reliable to trap all the moisture contained in the air and therefore to store ice samples with the same isotopic composition as the initial vapour (He and Smith, 1999; Schoch-Fischer et al., 1983; Steen-Larsen et al., 2011; Uemura et al., 2008). Two different cryogenic trapping setups have been deployed. The first one in 2006/07 was based on traps without glass balls. These traps cannot be used with air flow above 6 L/min in order to trap all the moisture because the surface available for thermal transfer is rather small. In order to be certain of trapping all the moisture, two traps in series were installed. Because of the lack of glass balls, the absence of water in the trap at the end of the

detrapping can be observed. This was a very important validation because detrapping efficiency is essential to obtain correct values of isotopic composition (Uemura et al., 2008). During the second campaign, we used traps filled with glass balls to increase the surface available for thermal transfer and therefore that can be used at higher flows. This cryogenic trapping setup relies on extensive tests previous to the campaign indicating that our custom-made glass traps filled with glass balls at -100°C successfully condensates all the moisture even for a flow up to 20L/min. These tests have been realised with 1. a Picarro (L2140i) to attest that the remaining humidity was below the measurement limit (around 30 ppmv) and 2. with a second trap downstream to evaluate the presence of ice after a period of 12 hours which would indicate a partial vapour trapping. These tests enable to validate the system we used, similar to Steen-Larsen et al. (2011), and motivate its deployment for the second campaign at Dome C. Extensive tests have also proven that complete detrapping can be done with traps filled with glass balls despite no direct observation of possible remaining water. The results shown later on (Figure 10) shows that similar values are obtained from both types of setup (with or without glass balls) and assess the reliability of both the methods.

Here, we present the results of two cryogenic trapping campaigns: one in 2006/07 and one in 2014/15."

———

"3) Calibration procedure for laser measurement Because the authors used custom manufactured calibration system, complete description for their calibration procedure is required. However, in its current form, it is very hard to understand their procedure except for expert of this field. For example, it was very hard to understand the following sentence: "Line 298: a series of calibration was performed in the laboratory from 100 to 1000 ppmv". Without the explanation of "a series of calibration", readers can't understand the meaning of this sentence. In section 2.4, there are several sentences similar to this. I recommend to reorganize and rewrite section 2.4 with the following information. 1. Schematic figure of self-designed calibration devise 2. Outline of the

calibration procedure (order of three type of calibrations and their frequency) 3. Operating procedure for each of calibration (don't forget the description for the SMOW-SLAP linearity correction) 4. Isotopic values of standard water for vapor sources 5. Accuracy and precision of this method (Laboratory experiment) 6. Calibration procedure for field campaign 7. Data quality of field data (include uncertainty of d-excess)"

The recommendation of the reviewer for a clearer calibration section has been followed. The calibration device, including schematic has been moved to supplementary material not to over charge the main manuscript. Please find here the entire section 2.4 with modifications of the text highlighted in red :

[revised manuscript text omitted]

* * *
All the minor concerns have been included and we are thankful for the reviewer to have taken the time to correct all these mistakes.

"Minor concerns P4, l167: figure 2 -> Figure 2 P6, l213: This noise -> These P6 l221: realized -> carried out? P6 l223: realized -> continued? P6 l237: figure 3 -> Figure 3 P7 l258-259: "Because we are working.." is the same meaning of the following sentence "L261-262: Standard calibration. . ." Please remove one of the other. P7 l265: Harvard Apparatus -> add parts number of this item P8 l268: Bronkhorst devices -> add parts number of it P12 l440: figure 6 -> Figure 6 P12 l459: figure 6 -> Figure 6 P13 l481: I think date expression is as follows: December 25th; 25th December; 25 December; December the twenty-fifth; the twenty-fifth of December I feel strange of the following expression: "December the 25th". Please check the expression of date. P16 l580: figure 9 -> Figure 9 P17 l619: I can't catch the meaning of "slope at the mid height". Please rephrase it. P16,l621: Figure 9 shows the negative peak of delta-18O at 18:00, corresponding to the minimum of surface temperature. Isn't this peak significant? P18,

[Figure]

Section 3.3: I can't understand why the authors stick to the discussion of slope value for the delta-D-deltaO plot. The slope is sensitive to the systematic bias so that the authors can't escape from the affect of large uncertainty of the measured isotopic values. As shown in Figure 10, I think that the most remarkable features of new data is extremely high d-excess values. So, I recommend to discuss the reason of these high values in here.

References Aemisegger, F., Sturm, P., Graf, P., Sodemann, H., Pfahl, S., Knohl, A., & Wernli, H. (2012). Measuring variations of 18O and 2H in atmospheric water vapour using two commercial laser-based spectrometers: an instrument characterisation study. Atmospheric Measurement Techniques, 5(7), 1491–1511. Bastrikov, V., Steen-Larsen, H. C., Masson-Delmotte, V., Gribanov, K., Cattani, O., Jouzel, J., & Zakharov, V. (2014). Continuous measurements of atmospheric water vapour isotopes in western Siberia (Kourovka). Atmospheric Measurement Techniques, 7(6), 1763–1776. He, H., & Smith, R. B. (1999). Stable isotope composition of water vapor in the atmospheric boundary layer above the forests of New England. Journal of Geophysical Research: Atmospheres, 104(D), 11657. Helliker, B. R., Roden, J. S., Cook, C., & Ehleringer, J. R. (2002). A rapid and precise method for sampling and determining the oxygen isotope ratio of atmospheric water vapor. Rapid Communications in Mass Spectrometry, 16(10), 929–932. Schoch-Fischer, H., K. Rozanski, H. Jacob, C. Sonntag, J. Jouzel, G. Ostlund, and M. Geyh. (1984). Hydrometeorologicalfactorscon- trolling the time variation of D,18 O and 3 H in atmospheric water vapour and precipitation in the northern westwind belt. Isotope Hydrology 1983, IAEA-publication, Vienna, Austria, 3–30 Steen-Larsen, H. C., Masson-Delmotte, V., Sjolte, J., Johnsen, S. J., Vinther, B. M., Breon, F. M., et al. (2011). Understanding the climatic signal in the water stable isotope records from the NEEM shallow firn/ice cores in northwest Greenland. Journal of Geophysical Research, 116(D6), D06108. Steen-Larsen, H. C., Sveinbjörnsdóttir, A. E., Peters, A. J., Masson-Delmotte, V., Guishard, M. P., Hsiao, G., et al. (2014). Climatic controls on water vapor deuterium excess in the marine boundary layer of the North Atlantic based on 500 days of in situ, continuous measurements. Atmospheric

[Figure]

Chemistry and Physics, 14(15), 7741–7756. Uemura, R., Matsui, Y., Yoshimura, K., Motoyama, H., & Yoshida, N. (2008). Evidence of deuterium excess in water vapor as an indicator of ocean surface conditions. Journal of Geophysical Research, 113(D19), D19114."

Please also note the supplement to this comment:
http://www.atmos-chem-phys-discuss.net/acp-2016-8/acp-2016-8-AC1-supplement.zip
* * *